# GAN-based Vertical Federated Learning for Label Protection

## Abstract

Split learning (splitNN) has emerged as a popular strategy for addressing the high computational costs and low modeling efficiency in Vertical Federated Learning (VFL). However, despite its popularity, vanilla splitNN lacks encryption protection, leaving it vulnerable to privacy leakage issues, especially Label Leakage from Gradients (LLG). Motivated by the LLG issue resulting from the use of labels during training, we propose the Generative Adversarial Federated Model (GAFM), a novel method designed specifically to enhance label privacy protection by integrating splitNN with Generative Adversarial Networks (GANs). GAFM leverages GANs to indirectly utilize label information by learning the label distribution rather than relying on explicit labels, thereby mitigating LLG. GAFM also employs an additional cross-entropy loss based on the noisy labels to further improve the prediction accuracy. Our ablation experiment demonstrates that the combination of GAN and the cross-entropy loss component is necessary to enable GAFM to mitigate LLG without significantly compromising the model utility. Empirical results on various datasets show that GAFM achieves a better and more robust trade-off between model utility and privacy compared to all baselines. In addition, we provide experimental justification to substantiate GAFM's superiority over splitNN, demonstrating that it offers enhanced label protection through gradient perturbation relative to splitNN. Codes of GAFM are available at https://anonymous.4open.science/r/Generative-Adversarial-Federated-Model-BFF7/.

## 1 Introduction

Federated learning trains algorithms across multiple decentralized remote devices or siloed data centers without sharing sensitive data. There are three types of federated learning depending on the data partitioning methods used: horizontal federated learning (HFL), vertical federated learning (VFL), and federated transfer learning (Yang et al., 2019). VFL partitions the data vertically, where local participants have datasets with the same sample IDs but different features (Hardy et al., 2017). With stricter data privacy regulations like CCPA1 (Pardau, 2018) and GDPR3 (Voigt & Von dem Bussche, 2017), VFL is a viable solution for enterprise-level data collaborations, as it facilitates collaborative training and privacy protection. However, VFL faces challenges in terms of high memory costs and processing time overheads, attributed to the complex cryptographic operations employed to provide strong privacy guarantees (Zhang et al., 2020), including additive homomorphic encryption (Hardy et al., 2017) and secure multi-party computation (Mohassel & Zhang, 2017), which are computationally intensive. To address these issues, split learning (Gupta & Raskar, 2018; Vepakomma et al., 2018; Abuadbba et al., 2020) has emerged as an efficient solution, allowing multiple participants to jointly train federated models without encrypting intermediate results, thus reducing computational costs. SplitNN (Ceballos et al., 2020), which applies the concept of split learning to neural networks, has been used successfully in the analysis of medical data (Poirot et al., 2019; Ha et al., 2021).

Split learning, while reducing computational costs, poses substantial privacy risks due to the absence of encryption protection for model privacy. One specific privacy risk is Label Leakage from Gradients (LLG) (Wainakh et al., 2022), in which gradients flowing from the label party to the non-label (only data) party can expose the label information (Erdogan et al., 2021; Zhu et al., 2019; Wainakh et al., 2021). LLG is susceptible to exploitation for stealing label information in binary classification and

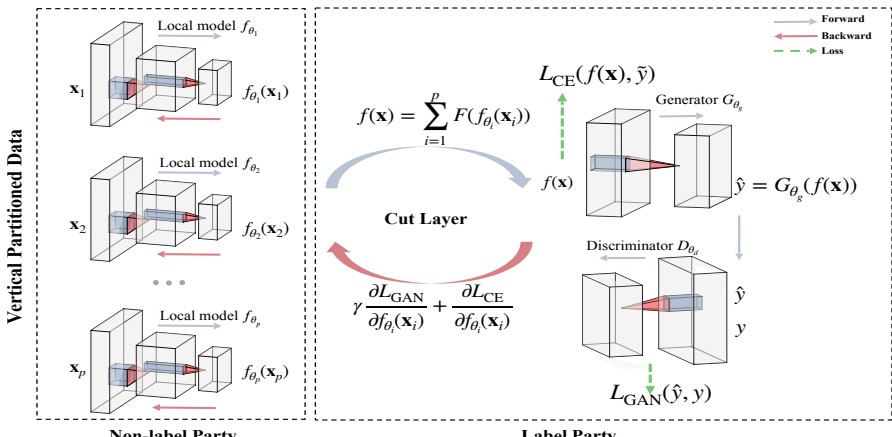

Figure 1: An illustration of Generative Adversarial Federated Model (GAFM) with $P$ non-label parties and one label party. Each non-label participant uses its local model $f_{\theta_p}$ to extract features $f_{\theta_p}(\mathbf{x}_p)$ from its local data $\mathbf{x}_p$, where $\theta_p$ is denoted as the local model parameter of the $pth$ non-label participant. The label party aggregates all feature information in the cut layer to obtain intermediate results $f(\mathbf{x})$, which are then used as input to the generator $G_{\theta_g}$ with $\theta_g$ being the generator model parameter. The label party then trains the generator $G_{\theta_g}$ and discriminator $D_{\theta_d}$ adversarially using the true label $y$ and the prediction $\hat{y}$ (the output of $G_{\theta_g}$), where $\theta_d$ is the discriminator model parameter. To incorporate additional true label information for improving the prediction accuracy, an additional cross-entropy loss $L_{\mathrm{CE}}$ is introduced. Section 3 provides more details on GAFM.

has limited proposed solutions to address it. Recent work by Li et al. (Li et al., 2021) indicates that *in binary classification, the gradient norm of positive instances is generally larger than negative ones*, which could potentially enable attackers to easily infer sample labels from intermediate gradients in splitNN. Despite the fact that binary classification is widely used in various federated scenarios, such as healthcare, finance, credit risk, and smart cities (Crowson et al., 2022; Byrd & Polychroniadou, 2020; Cheng et al., 2021; Zheng et al., 2022), and is vulnerable to LLG, limited research has been conducted on addressing the LLG issue in binary classification. Previous studies (Ceballos et al., 2020; Erdogan et al., 2021; Titcombe et al., 2021; Pereteanu et al., 2022) have focused mainly on securing the data information of non-label parties, while ignoring the risk of leaking highly sensitive label information of the label party. Therefore, it is critical to address how splitNN can resist LLG in binary classification tasks.

In this work, in order to prevent inferring sample labels from the gradient calculation, we introduce a novel Generative Adversarial Federated Model (GAFM), which synergistically combines the vanilla splitNN architecture with Generative Adversarial Networks (GANs) to indirectly incorporate labels into the model training process. Specifically, the GAN discriminator within GAFM allows federated participants to learn a prediction distribution that closely aligns with the label distribution, effectively circumventing the direct use of labels inherent in the vanilla SplitNN approach. Moreover, to counteract the potential degradation of model utility induced by GANs, we enhance our method by incorporating additional label information via an additional randomized cross-entropy loss, which encourages the intermediate results generated by the non-label party and the labels with added noise provided by the label party to perform similarly. The entire framework of the proposed GAFM and the training procedures is displayed in Figure 1. Our contributions are highlighted as follows.

- We propose a novel GAN-based approach, called GAFM, which combines vanilla splitNN with GAN to mitigate LLG in binary classification (section 3.2). Our analysis in section 3.4 demonstrates that GAFM protects label privacy by generating more mixed intermediate gradients through the mutual gradient perturbation of both the GAN and cross-entropy components.

- We enrich the existing gradient-based label stealing attacks by identifying two additional simple yet practical attack methods, namely mean attack and median attack in section 3.5. The experimental results in section 4.2.1 demonstrate that our new attacks are more effective in inferring labels than the existing ones.

- We evaluate the effectiveness of GAFM on various datasets. Empirical results in section 4 show that GAFM mitigates LLG without significant model utility degradation and the performance of GAFM across different random seeds is more stable compared to baselines. We also provide additional insights based on the ablation experiment (section 4.2.2) to demonstrate the benefit of combining both the GAN and cross-entropy components in GAFM.

## 2 RELATED WORK

**SplitNN-driven Vertical Partitioning.** SplitNN enables multiple participants to train a distributed model without sharing their data and encrypting intermediate results (Gupta & Raskar, 2018; Vepakomma et al., 2018; Ceballos et al., 2020). In SplitNN, each passive participant trains a partial neural network locally, and the layer at which the label and non-label participants share information is called the cut layer. At the cut layer of SplitNN, each non-label party trains a fixed portion of the neural network locally and shares intermediate results with the label party. The label party then aggregates these intermediate results and implements backward propagation to update the local parameters of each non-label party. There are several methods of aggregation, such as element-wise average, element-wise maximum, element-wise sum, element-wise multiplication, concatenation, and non-linear transformation (Ceballos et al., 2020).

**Label privacy protection via random gradient perturbation.** To address the issue of LLG through intermediate gradients at the cut layer, one possible solution is to introduce randomness to the intermediate gradients, which has been utilized in HFL (Abadi et al., 2016; Geyer et al., 2017; Hu et al., 2020). **Marvell** (Li et al., 2021) is a random perturbation approach specifically designed for binary classification tasks. Marvell protects label privacy by perturbing the intermediate gradients and aims to find the optimal zero-centered Gaussian perturbations that minimize the sum of KL divergences between two perturbed distributions, while adhering to a budget constraint on the amount of perturbation added:

$$\min_{W^{(0)}, W^{(1)}} \mathrm{KL}(\tilde{\mathbb{P}}^1 \| \tilde{\mathbb{P}}^0) + \mathrm{KL}(\tilde{\mathbb{P}}^0 \| \tilde{\mathbb{P}}^1)$$
$$\text{s.t.} \quad \mathrm{ptr}(\Sigma_0) + (1-\mathrm{p})\mathrm{tr}(\Sigma_1) \leq \mathrm{P}, \tag{1}$$

where $\tilde{\mathbb{P}}^k$ is the distribution of perturbed intermediate gradients from $k$ after convolution with the Gaussian noise $W^{(k)} = \mathcal{N}(0, \boldsymbol{\Sigma}_k)$, $k = 0, 1$, p is the weight and P is the budget for how much random perturbation Marvell is allowed to add. As the intermediate gradients with different labels become less distinguishable, it becomes more difficult for attackers to infer labels from gradients.

**Max Norm** (Li et al., 2021) is an improved heuristic approach of adding zero-mean Gaussian noise with non-isotropic and example-dependent covariance. More concretely, for the intermediate gradient $\mathbf{g}_j$ of data point $j$, Max Norm adds the zero-mean Gaussian noise $\eta_j$ to it with the covariance as:

$$\sigma_j = \sqrt{\frac{\|\mathbf{g}_{\max}\|_2^2}{\|\mathbf{g}_j\|_2^2} - 1}, \tag{2}$$

where $\|\mathbf{g}_{\max}\|_2^2$ is the largest squared 2-norm in a batch. Max Norm is a simple, straightforward, and parameter-free perturbation method. But it does not have strong theoretical motivation and cannot guarantee to defend unknown attacks (Li et al., 2021).

**GAN in Federated Learning.** Previously, researchers have combined GAN and HFL for various purposes, including three directions in HFL: (1) generating malicious attacks (Hitaj et al., 2017; Wang et al., 2019), (2) training high-quality GAN across distributed data under privacy constraints (Hardy et al., 2019; Rasouli et al., 2020; Mugunthan et al., 2021), and (3) protecting client data privacy (Wu et al., 2021), known as FedCG. In FedCG, each client train a classifier predicting the response $y$ using $Z$ extracted from the original feature $X$, and a conditional GAN that learns the conditional distribution of $Z$ given $y$. The classifiers and generators (instead of the extracted $Z$) from different clients are then passed to the server for updating the common model. FedCG has been shown to effectively protect clients' data privacy in HFL. The preceding studies have showcased the promising utility of GANs in Horizontal Federated Learning (HFL). However, as far as our awareness extends, there exists no prior investigation into the application of GAN models for label protection in Vertical

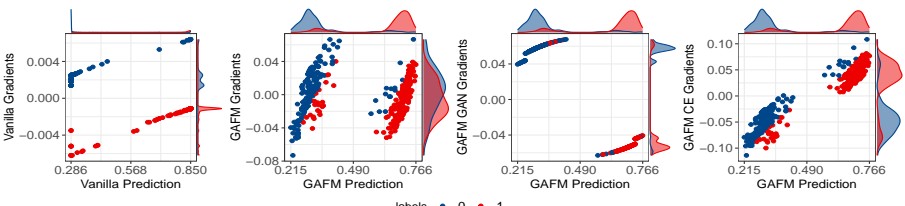

Figure 2: **Comparison of Prediction vs. Intermediate Gradients** between vanilla splitNN and GAFM on IMDB. The figure 2 displays, from left to right, the intermediate gradients of vanilla splitNN, GAFM, the GAN loss gradient of GAFM, and the CE loss gradient of GAFM. Our observations show that (1) GAFM has more mixed intermediate gradients compared to vanilla splitNN; (2) the gradient class centers from the GAN loss gradient and the CE loss gradient differ in opposite directions, leading to mutual perturbation in the final gradient for GAFM. Observations from Spambase, Criteo, and ISIC in Appendix are consistent with those from the IMDB dataset.

Federated Learning (VFL), where multiple participating parties jointly learn label distributions to mitigate the privacy risks arising from direct label usage.

## 3 GENERATIVE ADVERSARIAL FEDERATED MODEL (GAFM)

In this section, we first introduce the split learning problem for binary classification, including its associated notations. We then provide a detailed description of the GAFM method, along with a discussion on GAFM's parameter selection. Additionally, we explain how GAFM offers better label privacy protection than vanilla splitNN and propose new label stealing methods to evaluate GAFM's privacy protection ability in subsequent experiments.

### 3.1 PROBLEM SETTING AND NOTATION

We consider the joint training of one label party and $P$ non-label parties for binary classification tasks over the domain $\mathcal{X} \times \{0, 1\}$. Each non-label party $p$ possesses one local model $f_{\theta_p} : \mathcal{X} \to \mathbb{R}^d$ where $\theta_p$ is the local model parameter of non-label party $p$ and the local data $\mathbf{x}_p \in \mathcal{X}$. In vanilla splitNN, the label party owns the transformation function $F : \mathbb{R}^d \to \mathbb{R}^d$ for feature aggregation in the cut layer, the logit function $h : \mathbb{R}^d \to \mathbb{R}$ for prediction, and each example's label $y \in \{0, 1\}$. In contrast, in GAFM, the transformation function $F$ for feature aggregation remains with the label party, while the logit function $h$ is replaced by the generator $G_{\theta_g} : \mathbb{R}^d \to \mathbb{R}$ where $\theta_g$ is the generator model parameter. Furthermore, GAFM includes an additional discriminator $D_{\theta_d} : \mathbb{R} \to \mathbb{R}$ where $\theta_d$ is the discriminator model parameter.

### 3.2 GAFM FRAMEWORK

In this section, we provide a detailed description of the GAFM framework. As illustrated in Figure 1, each non-label party $p$ in GAFM utilizes its local model $f_{\theta_p}$ to extract local feature information from its local data $\mathbf{x}_p$ and sends it to the label party. In the cut layer, the label party aggregates the feature information from all non-label parties to obtain the intermediate result $f(\mathbf{x}) = F(\sum_p f_{\theta_p}(\mathbf{x}_p))$, where $F(.)$ is a known transformation function, such as the identity function. The label party takes the intermediate result $f(\mathbf{x})$ as input to the generator $G_{\theta_g}$, and the output of $G_{\theta_g}$ is treated as the prediction $\hat{y}$. Then, the label party adversarially trains both the generator $G_{\theta_g}$ and the discriminator $D_{\theta_d}$ with guidance from the true label $y$ and the prediction $\hat{y}$. The GAN loss $L_{\text{GAN}}$ between $G_{\theta_g}$ and $D_{\theta_d}$ is defined in equation 3, which quantifies the distance between the prediction $\hat{y}$ and the ground truth $y$ distribution by employing the Wasserstein-1 distance (Kantorovich, 1960):

$$L_{\text{GAN}}(\theta_d, \theta_g; \varepsilon) = E_y \left[ D_{\theta_d}(y + \varepsilon) \right] - E_{\mathbf{x}} \left[ D_{\theta_d}(\hat{y}) \right], \tag{3}$$

where $\hat{y} = D_{\theta_g}(f(\mathbf{x}))$ and $\varepsilon \sim N(0, \sigma^2)$ is a small additive Gaussian noise so that $y + \varepsilon$ has a continuous support. The Wasserstein loss, which encourages the predicted distribution to closely align with the empirical response, is applicable to classification tasks (Frogner et al., 2015).

To improve the prediction accuracy in GAFM, we introduce an additional cross-entropy (CE) loss, which measures the distance between the intermediate result $f(\mathbf{x})$ and the randomized response $\tilde{y}$. The randomized response $\tilde{y}$ is defined as:

$$\tilde{y} = \begin{cases} 0.5 + u; y = 1 \\ 0.5 - u; y = 0, \end{cases} \quad (4)$$

where the random variable $u \sim \mathrm{Uniform}(0, \Delta)$ and the hyperparameter $\Delta \in [0, 0.5]$. The randomized response $\tilde{y}$ replaces the original response $y$, which enhances label protection against LLG. Setting $\Delta = 0$ removes any label information from the original data, while $\Delta > 0$ allows $\tilde{y}$ to guide the classifier without excessive label leakage from CE loss. The use of a randomized response is a departure from previous work, which used the true label $y$ and encountered issues with label leakage. Using the randomized response in equation 4, the CE loss is:

$$L_{\mathrm{CE}}(\mathbf{\Theta}; u) = -E_{\mathbf{x}y}[\tilde{y}\log(\sigma(f(\mathbf{x}))) + (1 - \tilde{y})\log(\sigma(1 - f(\mathbf{x}))]. \quad (5)$$

where $\mathbf{\Theta} = \{\theta_1, \ldots, \theta_P\}$ is the local model parameter set and the function $\sigma$ is the sigmoid function. In particular, $\tilde{y} \in \mathbb{R}$ and $f(\mathbf{x}) \in \mathbb{R}^d$, where $d$ is an arbitrary dimension. When $d \neq 1$, $\sigma(f(\mathbf{x}))$ needs to be averaged along the $d$-dimension to be consistent with the dimension of $\tilde{y}$.

Combine the $L_{\mathrm{GAN}}$ defined in equation 3 and the $L_{\mathrm{CE}}$ defined in equation 5, we obtain the full loss of GAFM, as given by equation (6):

$$\min_{\theta_g, \theta_p} \max_{\theta_d} L_{\mathrm{GAFM}}(\theta_d, \theta_g, \mathbf{\Theta}; \varepsilon, u, \gamma) = \gamma L_{\mathrm{GAN}}(\theta_d, \theta_g; \varepsilon) + L_{\mathrm{CE}}(\mathbf{\Theta}; u), \quad (6)$$

where the loss weight hyperparameter $\gamma > 0$.

Following the parameter update sequence of Wasserstein-based GANs (Arjovsky et al., 2017a), GAFM enables the label party to update the discriminator parameter $\theta_d$ first, followed by updating the generator parameter $\theta_g$. Then, the non-label parties update local model parameters $\mathbf{\Theta}$:

(1) Update $\theta_d$ to maximize the GAN loss: $\theta_d \leftarrow \arg\max_{\theta_d} L_{\mathrm{GAN}}(\theta_d, \theta_g; \varepsilon)$, given $\theta_g$, $\mathbf{\Theta}$.

(2) Update $\theta_g$ to minimize the GAN loss: $\theta_g \leftarrow \arg\min_{\theta_g} L_{\mathrm{GAN}}(\theta_d, \theta_g; \varepsilon)$, given $\theta_d$, $\mathbf{\Theta}$.

(3) Update $\mathbf{\Theta}$ to minimize GAFM loss: $\mathbf{\Theta} \leftarrow \arg\min_{\mathbf{\Theta}} L_{\mathrm{GAFM}}(\theta_d, \theta_g, \mathbf{\Theta}; \varepsilon, u)$, given $\theta_d$, $\theta_g$.

We employ the Adam optimizer with learning rates $\alpha_d$, $\alpha_g$, and $\alpha_p$, respectively, to implement the aforementioned procedures. To satisfy the Lipschitz constraint of the discriminator in Wasserstein-based GANs (Arjovsky et al., 2017b; Gulrajani et al., 2017), we apply weight clipping to discriminator parameters $\theta_d$. This operation restricts the parameters $\theta_d$ to the interval $[-c, c]$, where $c$ is a hyperparameter. Further details regarding the algorithm can be found in Algorithm 1.

### 3.3 SELECTION OF GAFM HYPERPARAMETERS

GAFM incorporates three crucial parameters: $\sigma$, $\gamma$, and $\Delta$. The parameter $\sigma$ introduces Gaussian noise $\varepsilon \sim N(0, \sigma^2)$ into the ground truth $y$ distribution to render it continuously supported. The default value of $\sigma$ can be set to 0.01, and the appendix D.1 shows that GAFM is insensitive to different values of $\sigma$. Specifically, a broad range of $\sigma \in [0.01, 1]$ yields comparable empirical results.

The tuning of the loss weight hyperparameter $\gamma$ in GAFM can be challenging due to the vanishing gradient issue of the unnormalized GAN loss gradients (Arjovsky et al., 2017b; Gulrajani et al., 2017). To address this issue, we normalize the gradients of the GAN loss and the CE loss, respectively. This normalization alleviates the vanishing gradient issue, making it more feasible to determine the hyperparameter $\gamma$. Appendix D.2 provides details on training $\gamma$ for different datasets where experiments demonstrate that an effective approach to determine $\gamma$ is to adjust it to balance the magnitude between the average value of the GAN loss gradient and the average value of the CE loss gradient.

We use a simple yet effective approach for determining the randomized response-related hyperparameter $\Delta$. Specifically, we train GAFM with different values of $\Delta$ on a 10% subset of the data and select the optimal parameter based on its utility and privacy. The analysis in the appendix shows that the $\Delta$ selected based on the subset is very close to or even equal to the optimal parameter selected on the full dataset. Additional details on the selection of $\Delta$ are discussed in the appendix D.3.

### 3.4 WHY DOES GAFM PROTECT AGAINST LLG?

In this section, we show that GAFM provides stronger privacy protection against LLG by generating more mixed intermediate gradients compared to vanilla splitNN. Specifically, the proposition 3.1 from the work (Li et al., 2021) states that the KL divergence between the intermediate gradients of two classes sets an upper bound on the amount of label information an attacker can obtain via any LLG attacks.

**Proposition 3.1.** *(Li et al., 2021) Let $\tilde{\mathbb{P}}^1$ and $\tilde{\mathbb{P}}^0$ be perturbed distributions for intermediate gradients of classes 1 and 0, and be continuous with respect to each other. For $\epsilon \in [0, 4)$,*

$$\mathrm{KL}(\tilde{\mathbb{P}}^1 \| \tilde{\mathbb{P}}^0) + \mathrm{KL}(\tilde{\mathbb{P}}^0 \| \tilde{\mathbb{P}}^1) \leq \epsilon$$
$$implies \quad \max_r \mathrm{AUC_r} \leq \frac{1}{2} + \frac{\sqrt{\epsilon}}{2} - \frac{\epsilon}{8}, \tag{7}$$

*where $r$ is any LLG attack, and $\mathrm{AUC_r}$ represents the achieved AUC using $r$.*

The proposition 3.1 states that models with more mixed intermediate gradients (meaning a smaller sum of KL divergences from the two classes) imply a smaller upper bound $\epsilon$ for worst case leak AUC against LLG. We indirectly demonstrate GAFM's superior label privacy protection ability by showing that GAFM has more mixed intermediate gradients than vanilla splitNN. Figure 2 illustrates the intermediate gradients of vanilla splitNN, GAFM, the GAN loss gradient of GAFM, and the CE loss gradient of GAFM on the IMDB dataset from left to right. We observe that GAFM has more mixed intermediate gradients compared to vanilla splitNN. Furthermore, the gradient class centers of the GAN loss gradient and the CE loss gradient differ in opposite directions, leading to mutual cancelation in the final gradient for GAFM. We further provide a justification in Appendix B to explain the observed discrepancy in the directions of the GAN loss gradient class centers and the CE loss gradient class centers shown in Figure 2.

### 3.5 LABEL STEALING ATTACKS

In this section, we introduce three gradient-based label stealing attacks which are applied in subsequent experiments to evaluate the effectiveness of GAFM in mitigating LLG.

**Norm Attack.** The norm attack (Li et al., 2021) is a simple heuristic for black-box attacks that can be used for label inference in binary classification tasks. The attack is based on the observation that the gradient norm $\|\mathbf{g}\|_2$ of positive instances tends to differ from that of negative instances, especially with unbalanced datasets. Thus, the gradient norm $\|\mathbf{g}\|_2$ can be a strong predictor of labels.

Previous research indicates that vanilla SplitNN gradients tend to form two clusters based on the cosine similarity sign, leading to the development of cosine attacks (Li et al., 2021). In the case of GAFM, although the relationship between the sign and the cluster of gradients may not be as straightforward, attackers can still attempt to make attacks based on proximity to a cluster. To achieve this goal, we propose mean-based attacks and further enhance robustness against outliers in gradients by introducing median-based attacks.

**Mean Attack.** We propose a mean-based attack as a heuristic for black-box attacks that exploits clustering structures in gradients. Assuming that attackers know the gradient centers of class 0 and class 1, denoted as $\boldsymbol{\mu}_0$ and $\boldsymbol{\mu}_1$, mean-based attacks assign a sample $i$ to the cluster that its intermediate gradient $\mathbf{g}_j$ is closer to:

$$y_i = \begin{cases} 1, & \text{if } \|\mathbf{g}_i - \boldsymbol{\mu}_1\|_2 \leq \|\mathbf{g}_i - \boldsymbol{\mu}_0\|_2 \\ 0, & \text{otherwise.} \end{cases} \tag{8}$$

**Median Attack.** We propose a median-based attack, which is similar to the mean-based attack but more robust to outliers. Assuming that attackers know the gradient medians of class 0 and class 1, denoted as $\mathbf{m}_0$ and $\mathbf{m}_1$, median-based attacks assign a sample $i$ to the cluster that its intermediate gradient $\mathbf{g}_j$ is closer to:

$$y_i = \begin{cases} 1, & \text{if } \|\mathbf{g}_i - \mathbf{m}_1\|_2 \leq \|\mathbf{g}_i - \mathbf{m}_0\|_2 \\ 0, & \text{otherwise.} \end{cases} \tag{9}$$

Table 1: **Dataset statistics and model architectures.** Different local model architectures $f_{\Theta}$ are considered for different datasets. 1-layer DNNs are sufficient as the generator $G_{\theta_g}$ and discriminator $D_{\theta_d}$ since their inputs are simple linear embeddings.

| Dataset | Positive Instance Proportion | $f_{\Theta}$ |
|---------|------------------------------|--------------|
| Spambase | 39.90% | 2-layer DNN |
| IMDB | 50.00% | 3-layer DNN |
| Criteo | 22.66% | WideDeep & Model |
| ISIC | 1.76% | 6-layer CNN |

## 4 EXPERIMENTS

In this section, we first provide a comprehensive description of the experimental setup, including the datasets, model architectures, baselines, and evaluation metrics. Then, we present the experimental results that showcase the utility and privacy performance of GAFM . Furthermore, we conduct an ablation study to elucidate the contribution of each component of GAFM.

### 4.1 EXPERIMENT SETUP

**Datasets.** Our approach is evaluated on four real-world binary classification datasets: Spambase [1], which is used for spam email discrimination; IMDB[2], a binary sentiment classification dataset consisting of 50,000 highly polar movie reviews; Criteo [3], an online advertising prediction dataset with millions of examples; and ISIC [4], a healthcare image dataset for skin cancer prediction. Criteo and ISIC are two datasets with severely imbalanced label distributions, which have been used in the works of Max Norm and Marvell (Li et al., 2021). Furthermore, we consider another two class-balanced datasets, Spambase and IMDB, to evaluate the performance of GAFM under different class setting. Additional information regarding the datasets and data preprocessing can be found in Table 1 and the appendix part.

**Model Architecture.** We consider the two-party split learning setting used by Marvell, where the label party has access only to the label and the non-label party has access only to the data. In the appendix section, we also present the utility and privacy protection results of all methods under a multi-client scenario with three non-label parties. Table 1 lists the model architectures of local models $f_{\Theta}$, generator $G_{\theta_g}$, and discriminator $D_{\theta_d}$ used for each dataset. Similar to Marvel, we employ a Wide & Deep model (Cheng et al., 2016) for Criteo and a 6-layer CNN for ISIC as the local models $f_{\Theta}$. More details on model architectures and training can be found in appendices.

**Baselines.** We compare the utility and privacy of GAFM with Marvell (Li et al., 2021), Max Norm (Li et al., 2021) and vanilla splitNN. Both Marvell and MaxNorm are novel approaches designed to tackle the LLG challenge observed in binary classification within VFL.

**Evaluation Metrics.** We employ the Area Under Curve (AUC) metric to evaluate model utility and the leak AUC to assess model privacy protection. Differential privacy is not considered as a leakage measure, as it is not applicable to example-specific and example-aware settings like VFL (Li et al., 2021). The leak AUC (Li et al., 2021; Yang et al., 2022; Sun et al., 2022) is defined as the AUC achieved by using specific attacks. A high leak AUC value, closer to 1, indicates that the attacker can accurately recover labels, while a low leak AUC value, around 0.5, suggests that the attacker has less information for inferring labels. To prevent simple label flipping from resulting in a higher leak AUC, we modify the leak AUC as shown in equation 10 for a predefined attack by flipping the label assignment if doing so results in a higher AUC:

$$\text{leakAUC} \leftarrow \max(\text{leakAUC}, 1 - \text{leakAUC}) \tag{10}$$

---

[1] https://archive.ics.uci.edu/ml/datasets/spambase
[2] https://www.kaggle.com/datasets/uciml/default-of-credit-card-clients-dataset
[3] https://www.kaggle.com/c/criteo-display-ad-challenge
[4] https://www.kaggle.com/datasets/nodoubttome/skin-cancer9-classesisic

Table 2: Comparison of utility (AUC) and privacy (leak AUC) between GAFM and baselines. Compared to Vanilla and Max Norm, GAFM achieves lower leak AUC at similar AUC, demonstrating its effectiveness in protecting label privacy. Compared to Marvell, GAFM demonstrates comparable and even better utility and privacy protection on most datasets, while also exhibiting greater stability across different random seeds. Table 2 illustrates that GAFM offers a better and more stable trade-off between utility and privacy compared to all baselines.

| Dataset | Method | Utility | | | Privacy | | |
|---|---|---|---|---|---|---|---|
| | | Avg. | Worst | Best | Norm Attack | Mean Attack | Median Attack |
| Spambase | GAFM | 0.93 | 0.91 | 0.95 | 0.56±0.04 | 0.67±0.05 | 0.66±0.05 |
| | Marvell | 0.71 | 0.59 | 0.82 | 0.53±0.02 | 0.70±0.01 | 0.70±0.01 |
| | Max Norm | 0.95 | 0.95 | 0.95 | 0.83±0.11 | 1.00±0.00 | 0.91±0.00 |
| | Vanilla | 0.95 | 0.95 | 0.96 | 0.85±0.07 | 1.00±0.00 | 0.91±0.00 |
| IMDB | GAFM | 0.88 | 0.88 | 0.89 | 0.52±0.01 | 0.60±0.01 | 0.60±0.01 |
| | Marvell | 0.80 | 0.71 | 0.89 | 0.52±0.01 | 0.73±0.01 | 0.73±0.01 |
| | Max Norm | 0.89 | 0.89 | 0.90 | 0.82±0.09 | 1.00±0.00 | 0.99±0.01 |
| | Vanilla | 0.89 | 0.89 | 0.90 | 0.82±0.09 | 1.00±0.00 | 0.99±0.01 |
| Criteo | GAFM | 0.67 | 0.64 | 0.73 | 0.68±0.06 | 0.80±0.09 | 0.77±0.05 |
| | Marvell | 0.70 | 0.65 | 0.76 | 0.76±0.08 | 0.86±0.08 | 0.78±0.04 |
| | Max Norm | 0.69 | 0.65 | 0.72 | 0.92±0.01 | 0.83±0.09 | 0.82±0.00 |
| | Vanilla | 0.72 | 0.69 | 0.77 | 0.92±0.06 | 0.91±0.11 | 0.82±0.00 |
| ISIC | GAFM | 0.68 | 0.66 | 0.69 | 0.62±0.09 | 0.66±0.15 | 0.68±0.11 |
| | Marvell | 0.64 | 0.51 | 0.69 | 0.65±0.04 | 0.69±0.01 | 0.66±0.01 |
| | Max Norm | 0.76 | 0.72 | 0.82 | 0.99±0.01 | 1.00±0.00 | 0.77±0.00 |
| | Vanilla | 0.77 | 0.73 | 0.82 | 0.99±0.01 | 1.00±0.00 | 0.78±0.00 |

## 4.2 RESULTS

This section presents AUC and leak AUC to demonstrate the trade-off between privacy and utility of GAFM. We also report the results of the ablation study, which illustrates the contribution of each component of GAFM. Each method is executed 10 times with unique random seeds and train-test splits. Further experiment details can be found in the appendix E.

### 4.2.1 EVALUATION OF UTILITY AND PRIVACY

Table 2 illustrates the average AUC and average leak AUC across four datasets. GAFM achieves comparable classification AUC with Vanilla and Max Norm, but demonstrates lower leak AUC, indicating its effectiveness in protecting label privacy and defending against LLG. Compared to Marvell, GAFM achieves considerably higher average AUC on Spambase and IMDB, and comparable AUC on Criteo and ISIC, while achieving comparable (slightly smaller) leak AUC. Importantly, GAFM demonstrates more stable performance than Marvell, with reduced variance between the worst and best AUC across different random seeds. This could be attributed to Marvell's gradient perturbation technique, which introduces Gaussian noise to intermediate gradients and can lead to unstable performance, as evidenced by the highly fluctuating leak AUC of Criteo and ISIC shown in the Marvell paper (Li et al., 2021). Additionally, we observe that our proposed mean attack and median attack achieve higher leak AUC on all four datasets compared to the norm attack. This indicates that mean attack and median attack are more effective gradient-based attacks for label stealing compared to the existing norm attack.

### 4.2.2 ABLATION STUDY

Figure 3 compares GAFM with two ablated versions: GAFM with only GAN loss (GAN-only) and GAFM with only CE loss (CE-only), under more severe mean and median attacks. The results highlight GAFM's superior balance between utility and privacy across diverse datasets. The CE-only model fluctuates significantly in privacy protection, and the GAN-only model consistently underperforms in terms of model utility. In particular, the CE-only model shows elevated leak AUCs

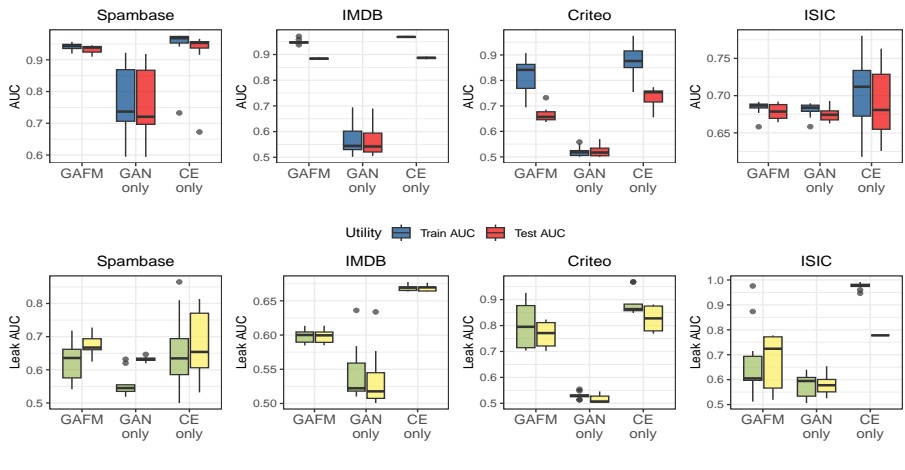

Figure 3: **Comparison of GAFM with two ablated versions**: GAN-only (GAFM with only GAN loss) and CE-only (GAFM with only CE loss). Figure 3 reveals that GAFM offers better trade-off between utility and privacy compared to the GAN-only model and the CE-only model. The GAN-only model achieves low leak AUC but has inferior classification AUC compared to GAFM. In contrast, the CE-only model fails to provide label privacy protection on most datasets. Figure 3 demonstrates the necessity of combining both the GAN and CE components.

Table 3: Comparison of GAFM with the CE-Only model in terms of the average ratio of AUC to leak AUC under Median and Mean Attack. GAFM strikes a better utility-privacy balance than the CE-Only model on most datasets, as evidenced by higher test AUC/leak AUC ratios.

| Dataset | Median Attack | | Mean Attack | |
|---|---|---|---|---|
| | GAFM | CE only | GAFM | CE only |
| Spambase | 1.39 | 1.39 | 1.50 | 1.47 |
| IMDB | 1.49 | 1.02 | 1.49 | 1.01 |
| Criteo | 0.87 | 0.89 | 0.84 | 0.83 |
| ISIC | 1.03 | 0.89 | 1.07 | 0.71 |

under both mean and median attacks on most datasets, notably reaching 0.7 or higher for IMDB and ISIC datasets. While the GAN-only model achieves low leak AUCs across all datasets, its classification AUC lags behind GAFM due to the guidance from the CE component. Table 3 presents the test AUC to leak AUC ratios under median and mean attacks, further emphasizing GAFM's superiority over the CE-Only model. This ablation study underscores the importance of synergizing GAN and CE components for robust LLG mitigation while preserving utility.

## 5 DISCUSSION

We propose GAFM, a novel method for binary classification tasks in VFL. Empirical experiments on four datasets demonstrate that GAFM is a promising method for effectively mitigating LLG. Unlike Marvell and Max Norm, which employ noisy intermediate gradients to enhance gradient mixing, GAFM adopts a unique approach. Incorporating GAN and CE losses, GAFM shows improved gradient mixing compared to vanilla splitNN. Our analysis indicates that GAFM's defense against LLG stems from mutual gradient perturbation of GAN and CE losses.

**Limitation and future work.** While heuristic explanations and extra experiments clarify how GAFM leverages mutually perturbing gradients from GAN and CE components for privacy protection, GAFM lacks the rigor of Marvell's upper bound on gradient-based label leakage. GAFM and Marvell are not exclusive; GAFM can integrate optimized noise from Marvell to enhance privacy. An example in the Appendix F highlights the combined benefits of GAFM and Marvell, outperforming individual methods in mitigating LLG at a slight utility cost. Additionally, extending these findings to multi-class settings and investigating unique features in such tasks is a promising avenue for future research.

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

## A    GAFM ALGORITHM DESCRIPTION AND ANONYMOUS CODE

The details of GAFM are outlined in Algorithm 1. We also provide an anonymized link that includes the GAFM code implementation.

---

**Algorithm 1** Generative Adversarial Federated Model

---

**Input**   : Training data $\mathbf{x}_1, \mathbf{x}_2, ..., \mathbf{x}_p$, training labels $y$, sample $u \sim Uniform(0, \Delta)$, $\varepsilon \sim N(0, \sigma^2)$, epoch $T$, weight $\gamma$, learning rates $\alpha_d, \alpha_g, \alpha_L$ and clip value $c$.

1 Initialize model parameters $\theta_d^0, \theta_g^0$ and $\theta_p^0$ and calculate the initialized intermediate result $f(\mathbf{x})$.

2 **while** *Not converge* **do**

3    **Label party**
      /* Step 1:   Update the discriminator                                */

4    The label party computes the prediction $\hat{y} \leftarrow G_{\theta_g}(f(\mathbf{x}))$ and updates the discriminator by
      $\theta_d \leftarrow \theta_d + \alpha_d \nabla_{\theta_d} L_{\text{GAN}}$ and clip $\theta_d \leftarrow clip(\theta_d, -c, c)$.
      /* Step 2:   Update the generator                                    */

5    Then the label party updates the generator by $\theta_g \leftarrow \theta_g - \alpha_g \nabla_{\theta_g} L_{\text{GAN}}$.
      /* Step 3:   Back-propagation                                        */

6    The label party back propagates normalized gradients with respect to $f(\mathbf{x})$ to the passive parties
      $grad_{f(\mathbf{x})} = \gamma \dfrac{\nabla_{f(\mathbf{x})} L_{\text{GAN}}}{\left\| \nabla_{f(\mathbf{x})} L_{\text{GAN}} \right\|_2} + \dfrac{\nabla_{f(\mathbf{x})} L_{\text{CE}}}{\left\| \nabla_{f(\mathbf{x})} L_{\text{CE}} \right\|_2}$

7    **Non-label parties**
      **for** $p = 1, \ldots, P$ **do**

8       /* Step 4:   Update local models                                    */

9       Based on the locally model and $grad_{f(\mathbf{x})}$, passive party $p$ updates $\theta_p$.

10   **in parallel**
      /* Step 5:   Forward-propagation                                    */

11   Update intermediate results $f(\mathbf{x}) \leftarrow \sum_{p=1}^{P} f_{\theta_p}(\mathbf{x}_p)$.

12 **end**

---

## B    HEURISTIC JUSTIFICATION FOR IMPROVED GRADIENTS MIXING

Recall that $\tilde{y} = 0.5 + 2(y - \frac{1}{2})u$ is the randomized response, $\hat{y} = G_{\theta_g}(f(x))$ is the encoded value, and $f(x)$ is the link function value at $x$ when predicting $\tilde{y}$ via a sigmoid transformation. Here, we provide heuristic justification for why the GAN and penalty component in GAFM tends to have opposite direction.

- We assume that $f(\mathbf{x})$ to be increase with $y$ since it tries to match $\tilde{y}$, which tends to increase with $y$ itself.

To optimize the equation (6), for the sample $i$ , the intermediate gradient is:

$$\frac{\partial L_{\text{GAFM}}}{\partial f(\mathbf{x}_i)} = \gamma \frac{\partial L_{\text{GAN}}}{\partial f(\mathbf{x}_i)} + \frac{\partial L_{\text{CE}}}{\partial f(\mathbf{x}_i)}. \tag{11}$$

For the first term of equation (11):

$$\frac{\partial L_{\text{GAN}}}{\partial f(\mathbf{x}_i)} = -\frac{\partial D_{\theta_d}(G_{\theta_g}(f(\mathbf{x}_i)))}{N \partial f(\mathbf{x}_i)} = -\frac{\partial D_{\theta_d}(\hat{y}_i)}{N \partial \hat{y}_i} \frac{\partial \hat{y}_i}{\partial f(\mathbf{x}_i)}. \tag{12}$$

- Consider the original GAN loss $L_{\text{GAN}}$ and approximate it roughly with Taylor expansion:

$$L_{\text{GAN}}(\theta_d, \theta_g; \varepsilon) = \mathbb{E}\left[D_{\theta_d}(y + \varepsilon)\right] - \mathbb{E}\left[D_{\theta^d}(G_{\theta_g}(f(\mathbf{x})))\right]$$
$$= \mathbb{E}_{y=1}[D_{\theta_d}(1 + \varepsilon) - D_{\theta_d}(G_{\theta_g}(f(\mathbf{x})))] + \mathbb{E}_{y=0}[D_{\theta_d}\varepsilon - D_{\theta_d}(f(\mathbf{x}))]$$
$$\approx \mathbb{E}_{y=1}[\nabla D_{\theta_d}(1)(1 + \varepsilon - \hat{y})] + \mathbb{E}_{y=0}[\nabla D_{\theta_d}(0)(\varepsilon - \hat{y})]$$
$$\approx \mathbb{E}_{y=1}[\nabla D_{\theta_d}(1)(1 - \hat{y})] - \mathbb{E}_{y=0}[\nabla D_{\theta_d}(0)\hat{y}]$$
$$= \nabla D_{\theta_d}(1)(1 - \mathbb{E}_{y=1}[\hat{y}]) - \nabla D_{\theta_d}(0)\mathbb{E}_{y=0}[\hat{y}].$$

We consider a under-training scenario where $\hat{y}$ tends to be in $(0, 1)$, with $\mathbb{E}_{y=1}[\hat{y}] < 1$ and $E_{y=0}[\hat{y}] > 0$.

In this case, in order to maximize the GAN loss, $\nabla D_{\theta_d}(1)$ tends to large and positive, and $\nabla D_{\theta_d}(0)$ tends to be small and negative. As a consequence, heuristically, $\nabla D_{\theta_d}(\hat{y})$ tends to be positive and large if $\hat{y}$ is large, whereas $\nabla D_{\theta_d}(\hat{y})$ tends to be small and negative if $\hat{y}$ is small. From now on, we assume this to be true.

- Let $\mathbf{x}_0$, $\mathbf{x}_1$ be samples from class 0 and class 1, and $\hat{y}_0$, $\hat{y}_1$ be their corresponding encoder value respectively. Given that $\nabla D_{\theta_d}(\hat{y})$ is monotonely non-decreasing in $\hat{y}$ and $f(\mathbf{x})$ is monotonely increasing with $y$, we have:

  1. When $\hat{y}$ is increasing with $f(\mathbf{x})$: since $f(\mathbf{x}_1) \gtrsim f(\mathbf{x}_0)$, we have $\hat{y}_1 \gtrsim \hat{y}_0$ and, consequently, $\nabla D_{\theta_d}(\hat{y}_1) \gtrsim \nabla D_{\theta_d}(\hat{y}_0)$, where $\gtrsim$ represents the concept of "tends to be greater". In this case, plug them into eq. (12), we have $\frac{\partial L_{\text{GAN}}}{\partial f(\mathbf{x}_0)} \gtrsim \frac{\partial L_{\text{GAN}}}{\partial f(\mathbf{x}_1)}$.

  2. When $\hat{y}$ is decreasing with $f(\mathbf{x})$: since $f(\mathbf{x}_1) \gtrsim f(\mathbf{x}_0)$, we have $\hat{y}_0 \gtrsim \hat{y}_1$ and, consequently, $\nabla D_{\theta_d}(\hat{y}_0) \gtrsim \nabla D_{\theta_d}(\hat{y}_1)$. In this case, plug them into eq. (12), we still have $\frac{\partial L_{\text{GAN}}}{\partial f(\mathbf{x}_0)} \gtrsim \frac{\partial L_{\text{GAN}}}{\partial f(\mathbf{x}_1)}$.

Combining them together, the distribution of GAN gradients at $y = 1$ tends to be left of the distribution at $y = 0$.

For the second term of equation (11):

$$\frac{\partial L_{\text{CE}}}{\partial f(\mathbf{x}_i)} = \frac{1}{N}\left(-\frac{\tilde{y}}{\sigma(f(\mathbf{x}_i))} + \frac{1 - \tilde{y}}{1 - \sigma(f(\mathbf{x}_i))}\right) = \frac{1}{N}\frac{\sigma(f(\mathbf{x}_i)) - \tilde{y}}{\sigma(f(\mathbf{x}_i))(1 - \sigma(f(\mathbf{x}_i)))}. \tag{13}$$

Without the GAN loss part, in a perfectly fitted model, we tend to have,

$$\begin{cases} E(\sigma(f(\mathbf{x}_i))|y_i = 1) = E(\tilde{y}_i|y_i = 1) = 0.5 + \frac{\Delta}{2} \\ E(\sigma(f(\mathbf{x}_i))|y_i = 0) = E(\tilde{y}_i|y_i = 0) = 0.5 - \frac{\Delta}{2} \end{cases} \tag{14}$$

In practice, an imperfect fit tends to have $E(\sigma(f(\mathbf{x}_i))|y_i = 1) < E(\tilde{y}_i|y_i = 1)$ and $E(\sigma(f(\mathbf{x}_i))|y_i = 0) > E(\tilde{y}_i|y_i = 0)$. In addition, with the GAN loss part, the larger gradient (usually) for $y = 0$ from the GAN loss drives $\sigma(f(\mathbf{x}_i))$ to decrease more at $y = 0$ compared to that from $y = 1$. Combining them together, heuristically, we tend to have,

$$\frac{\partial L_{\text{CE}}}{\partial \sigma(f(\mathbf{x}_i))} = \begin{cases} \frac{\sigma(f(\mathbf{x}_i)) - \tilde{y}}{N\sigma(f(\mathbf{x}_i))(1 - \sigma(f(\mathbf{x}_i)))} > 0 & y_i = 1 \\ \frac{\sigma(f(\mathbf{x}_i)) - \tilde{y}}{N\sigma(f(\mathbf{x}_i))(1 - \sigma(f(\mathbf{x}_i)))} < 0 & y_i = 0 \end{cases} . \tag{15}$$

Penalty gradients at $y = 1$ tend to be the right of the distribution at $y = 0$.

Finally, combining equations (12) and (15), the normalized final gradient for $L_{GAFM}$ is

$$\frac{\partial L_{\text{GAFM}}}{\partial f(\mathbf{x}_i)} = \begin{cases} \gamma f_{\text{Norm}}(-\nabla D_{\theta_d}(\hat{y}_i)\frac{\partial \hat{y}_i}{\partial f(\mathbf{x}_i)}) \downarrow + f_{\text{Norm}}(\frac{\sigma(f(\mathbf{x}_i)) - \tilde{y}_i}{N\sigma(f(\mathbf{x}_i))(1 - \sigma(f(\mathbf{x}_i)))}) \uparrow & y_i = 1 \\ \gamma f_{\text{Norm}}(-\nabla D_{\theta_d}(\hat{y}_i)\frac{\partial \hat{y}_i}{\partial f(\mathbf{x}_i)}) \uparrow + f_{\text{Norm}}(\frac{\sigma(f(\mathbf{x}_i)) - \tilde{y}_i}{N\sigma(f(\mathbf{x}_i))(1 - \sigma(f(\mathbf{x}_i)))}) \downarrow & y_i = 0 \end{cases} , \tag{16}$$

where the normalization operation is $f_{\text{Norm}}(\mathbf{x}) = \frac{\mathbf{x}}{||\mathbf{x}||_2}$ which does not change the order of gradients. This heuristic analysis suggests that the gradients from the GAN component and the penalty component tends to have opposite direction. Equation (16) demonstrates GAN loss gradient and penalty loss gradient help the total gradient to mix better by mutual perturbation.

## C   COMPLETE EXPERIMENTAL RESULTS

### C.1   INTERMEDIATE GRADIENTS ON ADDITIONAL DATASETS

Figures 4, 5, and 6 show the intermediate gradients of Spambase, Criteo, and ISIC datasets, respectively. Consistent with the results on the IMDB dataset, we observe that GAFM has more mixed gradients compared to Vanilla on these three datasets. These observation support the analysis in Appendix B that the mixed gradients of GAFM can be attributed to the mutual cancellation between GAN loss gradients and CE loss gradients.

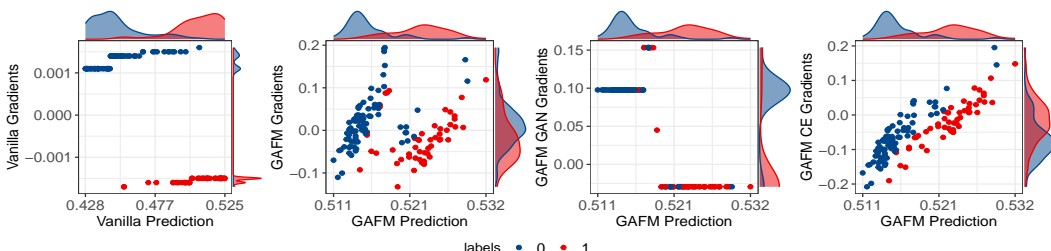

Figure 4: Comparison of Prediction vs. Intermediate Gradients between vanilla SplitNN and GAFM on Spambase. The figure displays, from left to right, the intermediate gradients of vanilla SplitNN, GAFM, the GAN loss gradient of GAFM, and the CE loss gradient of GAFM. We observe that the mutual perturbation between the GAN loss gradient and the CE loss gradient generate intermediate gradients that are more mixed compared to vanilla SplitNN.

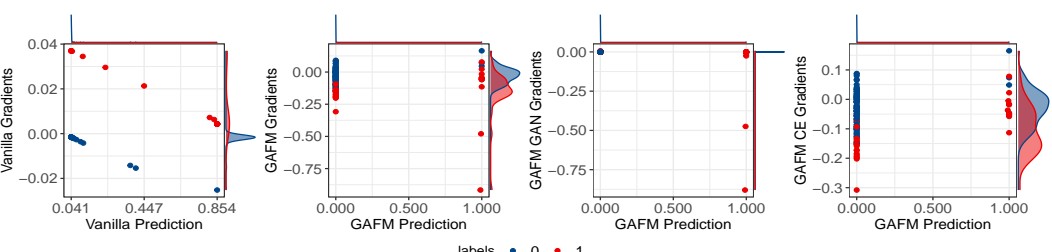

Figure 5: Comparison of Prediction vs. Intermediate Gradients between vanilla SplitNN and GAFM on Criteo. The figure displays, from left to right, the intermediate gradients of vanilla SplitNN, GAFM, the GAN loss gradient of GAFM, and the CE loss gradient of GAFM. We observe that the mutual perturbation between the GAN loss gradient and the CE loss gradient generate intermediate gradients that are more mixed compared to vanilla SplitNN.

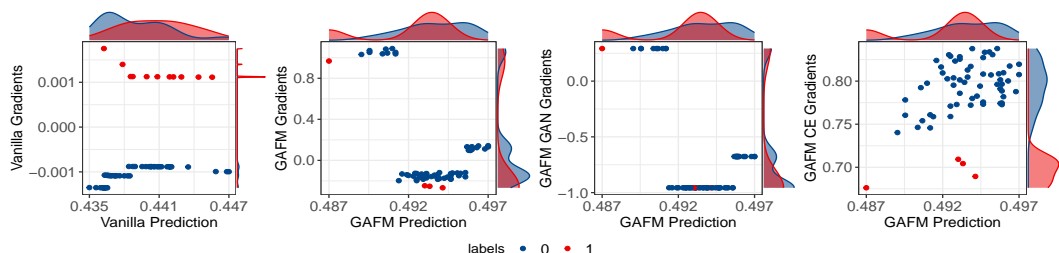

Figure 6: Comparison of Prediction vs. Intermediate Gradients between vanilla SplitNN and GAFM on ISIC. The figure displays, from left to right, the intermediate gradients of vanilla SplitNN, GAFM, the GAN loss gradient of GAFM, and the CE loss gradient of GAFM. We observe that the mutual perturbation between the GAN loss gradient and the CE loss gradient generate intermediate gradients that are more mixed compared to vanilla SplitNN.

## C.2 EVALUATION IN THE MULTI-PARTY SETTING

In this section, we extend the two-party split learning setting to the multi-party split learning setting, where three non-label parties and one label party collaborate to train GAFM. We evaluate two scenarios for local feature assignment: balanced and imbalanced. For Spambase and Criteo datasets, the local feature assignment ratio among the three participants is set to $client_1 : client_2 : client_3 = 1 : 1 : 1$, while for IMDB dataset, it is set to $client_1 : client_2 : client_3 = 2 : 2 : 1$. Since ISIC image dataset is not suitable for multi-party scenarios, we exclude it from our experiments. The experimental setup is consistent with that of section 4.1, with the exception of the transformation function $F(.)$. In this case, $F(.)$ is an averaging function rather than the identity function $I(.)$.

$$f(\mathbf{x}) = F(f(\mathbf{x}_1), f(\mathbf{x}_2), f(\mathbf{x}_3)) = \frac{f(\mathbf{x}_1) + f(\mathbf{x}_2) + f(\mathbf{x}_3)}{3}. \qquad (17)$$

where $f(\mathbf{x}_i)$ is the intermediate results from the $ith$ non-label party.

Table 4: Comparison of utility and privacy on multi-party setting over 10 repetitions. In the multi-party setting, GAFM still outperforms Vanilla and Max Norm in terms of privacy preservation, and Marvell in terms of utility.

| dataset | Method | Training AUC | Testing AUC | Norm Attack | Mean Attack | Median Attack |
|---------|--------|--------------|-------------|-------------|-------------|---------------|
| Spambase | GAFM | 0.87±0.15 | 0.86±0.15 | 0.60±0.07 | 0.69±0.06 | 0.68±0.07 |
| | Marvell | 0.70±0.05 | 0.68±0.06 | 0.53±0.03 | 0.70±0.02 | 0.70±0.02 |
| | Max Norm | 0.95±0.00 | 0.95±0.00 | 0.83±0.11 | 0.99±0.02 | 0.99±0.04 |
| | Vanilla | 0.95±0.00 | 0.95±0.00 | 0.84±0.13 | 1.00±0.00 | 1.00±0.00 |
| IMDB | GAFM | 0.97±0.01 | 0.82±0.00 | 0.54±0.02 | 0.65±0.04 | 0.54±0.02 |
| | Marvell | 0.82±0.01 | 0.78±0.01 | 0.51±0.01 | 0.71±0.02 | 0.71±0.02 |
| | Max Norm | 0.97±0.00 | 0.88±0.00 | 0.54±0.03 | 0.99±0.01 | 1.00±0.00 |
| | Vanilla | 0.97±0.00 | 0.88±0.00 | 0.54±0.02 | 0.99±0.01 | 1.00±0.00 |
| Criteo | GAFM | 0.80±0.01 | 0.65±0.02 | 0.63±0.01 | 0.86±0.01 | 0.80±0.00 |
| | Marvell | 0.83±0.03 | 0.70±0.04 | 0.75±0.08 | 0.85±0.09 | 0.78±0.04 |
| | Max Norm | 0.83±0.02 | 0.71±0.02 | 0.91±0.03 | 0.95±0.09 | 0.82±0.00 |
| | Vanilla | 0.83±0.02 | 0.71±0.02 | 0.91±0.03 | 0.95±0.09 | 0.82±0.00 |

The leak AUC in Tabel 4 measures not only the overall leakage, but also the leakage of each non-label party, as the $\tilde{y}_1$, $\tilde{y}_2$, and $\tilde{y}_3$ are equally weighted at the cut layer. We observe that the utility of both GAFM and Marvell slightly decreases in the multi-party setting compared to the two-party setting, possibly due to the increased difficulty of jointly training with multiple participants. However, in the multi-client setting, GAFM still achieves a better trade-off between utility and privacy compared to all baselines.

# D    DISCUSSION ON HYPERPARAMETERS

In this section, we present the methods for selecting the key hyperparameters $\sigma$, $\gamma$, and $\Delta$ in GAFM. Table 5 presents the hyperparameters of GAFM that are utilized in our experiments on four datasets.

Table 5: Hyperparameters of GAFM on four datasets.The following section discusses the selection of three crucial parameters in GAFM: $\sigma$ (section D.1), $\gamma$ (section D.2), and $\Delta$ (section D.3), respectively.

| Dataset | $\sigma$ | $\Delta$ | $\gamma$ |
|---|---|---|---|
| Spambase | 0.01 | 0.05 | 1 |
| IMDB | 0.01 | 0.1 | 1 |
| Criteo | 0.01 | 0.5 | 1 |
| ISIC | 0.01 | 0.05 | 20 |

## D.1    DISCUSSION ON $\sigma$

We conducted a comparison of GAFM with varying values of $\sigma$ over 10 repetitions, as shown in Table 6. Our results indicate that a moderately large $\sigma$ does not significantly compromise prediction accuracy and can still offer robust protection against label stealing attacks. Thus, we have chosen to fix $\sigma$ at 0.01 for all datasets.

Table 6: Average AUC and Leak AUC of GAFM with different $\sigma$. We observe that GAFM is insensitive to $\sigma$ in terms of both utility and privacy.

| dataset | $\sigma$ | Training AUC | Test AUC | Norm Attack | Mean Attack | Median Attack |
|---|---|---|---|---|---|---|
| Spambase | 0.01 | 0.94±0.01 | 0.93±0.02 | 0.56±0.04 | 0.67±0.05 | 0.66±0.05 |
| | 0.25 | 0.94±0.01 | 0.93±0.01 | 0.60±0.06 | 0.67±0.03 | 0.67±0.03 |
| | 1 | 0.94±0.01 | 0.94±0.01 | 0.57±0.05 | 0.68±0.04 | 0.68±0.04 |
| IMDB | 0.01 | 0.95±0.01 | 0.88±0.00 | 0.52±0.01 | 0.60±0.01 | 0.60±0.01 |
| | 0.25 | 0.92±0.14 | 0.85±0.11 | 0.52±0.02 | 0.60±0.03 | 0.61±0.03 |
| | 1 | 0.87±0.18 | 0.81±0.15 | 0.54±0.03 | 0.60±0.06 | 0.61±0.05 |
| Criteo | 0.01 | 0.82±0.07 | 0.67±0.03 | 0.68±0.06 | 0.80±0.09 | 0.77±0.05 |
| | 0.25 | 0.80±0.00 | 0.69±0.03 | 0.74±0.07 | 0.82±0.03 | 0.82±0.00 |
| | 1 | 0.81±0.02 | 0.66±0.03 | 0.71±0.08 | 0.82±0.02 | 0.82±0.00 |
| ISIC | 0.01 | 0.68±0.01 | 0.68±0.01 | 0.62±0.09 | 0.66±0.15 | 0.68±0.11 |
| | 0.25 | 0.68±0.01 | 0.67±0.01 | 0.63±0.06 | 0.69±0.13 | 0.69±0.09 |
| | 1 | 0.68±0.01 | 0.67±0.01 | 0.63±0.06 | 0.69±0.13 | 0.69±0.09 |

## D.2    DISCUSSION ON $\gamma$

We use the loss weight parameter $\gamma$ to balance the GAN and CE loss gradients in GAFM, allowing them to effectively perturb each other. Table 7 shows the average gradients of the GAN and CE losses for the GAFM model on four datasets, with $\gamma = 1$ and other hyperparameters consistent with the section 4.1. The results indicate that the mean gradients of the GAN and CE losses are similar in magnitude for the Spambase, IMDB, and Criteo datasets, but the CE loss gradient is over 10 times larger than the GAN loss gradient for the ISIC dataset. This explains why Spambase, IMDB, and Criteo can achieve satisfactory utility-privacy trade-offs with $\gamma = 1$, while ISIC requires a higher value of $\gamma$.

Table 7: Results of the average GAN loss gradient and average CE loss gradient on the four datasets. It is observed that only for the ISIC dataset, the CE loss gradient mean and GAN loss gradient mean are not on the same scale. To better achieve gradient perturbation, an increase in $\gamma$ for ISIC is necessary.

| dataset | $(\Delta, \gamma, \sigma)$ | GAN Gradient Avg. | CE Gradient Avg. | Comparable Magnitude |
|---|---|---|---|---|
| Spambase | (0.05,1,0.01) | 0.054 | -0.049 | ✓ |
| IMDB | (0.05,1,0.01) | 0.006 | -0.006 | ✓ |
| Criteo | (0.05,1,0.01) | -0.011 | 0.048 | ✓ |
| ISIC | (0.05,1,0.01) | 0.058 | 0.789 | ✗ |

We further discuss how to select the appropriate value of $\gamma$ for the ISIC dataset. Based on the results shown in Table 7, the value of $\gamma$ should ideally be in the range of 10 to 20. We set the parameter $\Delta$ to a minimum value of 0.05 to control its impact on privacy and utility while keeping all other parameters and experimental settings unchanged. Table 8 reports the experimental results of GAFM's privacy and utility under different values of $\gamma$. GAFM achieves comparable average training AUC and test AUC on the ISIC 10% subset across the range of $\gamma$ values. However, at $\gamma = 20$, GAFM achieves the minimum average leak AUC under three different attacks. Therefore, we conclude that the optimal parameter for the ISIC subset is $\gamma = 20$.

Table 8: The impact of different $\gamma$ values on the utility and privacy of GAFM on **the 10% subset ISIC data**. Table 8 reports the experimental results of GAFM with various $\gamma$ values, while maintaining all other experimental settings consistent. Moreover, we set the parameter $\Delta$ to a minimum value of 0.05 to control its effect on utility and privacy. Based on the overall assessment of utility and privacy, we conclude that the optimal value for $\gamma$ is 20.

| $\gamma$ | $(\Delta, \sigma)$ | Training AUC | Test AUC | Norm Attack | Mean Attack | Median Attack |
|---|---|---|---|---|---|---|
| 1 | (0.05,0.01) | 0.65±0.06 | 0.65±0.06 | 0.77±0.04 | 0.99±0.01 | 0.78±0.00 |
| 10 | (0.05,0.01) | 0.67±0.03 | 0.66±0.05 | 0.77±0.04 | 0.98±0.02 | 0.78±0.00 |
| 15 | (0.05,0.01) | 0.67±0.03 | 0.66±0.05 | 0.76±0.04 | 0.96±0.04 | 0.77±0.00 |
| 20 | (0.05,0.01) | 0.67±0.03 | 0.66±0.05 | 0.76±0.04 | 0.93±0.05 | 0.76±0.00 |

To validate the effectiveness of the selected $\gamma$ on the subset, we also report the experimental results on the full ISIC dataset with different values of $\gamma$ in Table 9. We find that $\gamma = 20$ is also an appropriate value for the full dataset. The consistency of the selected $\gamma$ for both the subset and the full dataset demonstrates the validity of determining $\gamma$ based on sampling.

Table 9: The impact of different $\gamma$ values on the utility and privacy of GAFM on **the full ISIC data**. Table 8 reports the experimental results of GAFM with various $\gamma$ values, while maintaining all other experimental settings consistent. Moreover, we set the parameter $\Delta$ to a minimum value of 0.05 to control its effect on utility and privacy. Based on the overall assessment of utility and privacy, we conclude that the optimal value for $\gamma$ is 20.

| $\gamma$ | $(\Delta, \sigma)$ | Training AUC | Test AUC | Norm Attack | Mean Attack | Median Attack |
|---|---|---|---|---|---|---|
| 1 | (0.05,0.01) | 0.67±0.02 | 0.66±0.03 | 0.98±0.01 | 0.97±0.02 | 0.78±0.00 |
| 10 | (0.05,0.01) | 0.67±0.02 | 0.68±0.01 | 0.77±0.19 | 0.80±0.12 | 0.74±0.04 |
| 15 | (0.05,0.01) | 0.67±0.04 | 0.66±0.05 | 0.61±0.07 | 0.66±0.15 | 0.68±0.09 |
| 20 | (0.05,0.01) | 0.68±0.01 | 0.68±0.01 | 0.62±0.09 | 0.66±0.15 | 0.68±0.11 |

## D.3 DISCUSSION ON $\Delta$

We now turn to the topic of determining the randomized response-related hyperparameter $\Delta$ in GAFM. In section 4.1, we chose $\Delta$ based on the measure $Ratio = \frac{leakAUC}{trainAUC}$ and the minimum average ratio criterion. Specifically, we selected $\Delta$ that yields the lowest average ratio across all three attacks.

We follow the procedure outlined below. After fixing the other parameters, we only randomly sample a small subset (10%) of the dataset without replacement to ensure privacy, which is shared between the label and non-label parties. GAFM is trained on this small subset with different values of $\Delta$, and we repeat this process 5 times. We then compute the ratio of norm attack, mean attack, and median attack for each value of $\Delta$ and average these ratios across the repetitions. The results are presented in Table 10. Based on these ratios, we select the optimal $\Delta$ for each dataset subset. Specifically, the optimal $\Delta$ for the Spambase subset is 0.05, for the IMDB subset is 0.1, for the Criteo subset is 0.5, and for the ISIC subset is 0.05.

To better demonstrate the effectiveness of the ratio-based strategy, we compare the detailed ratios on both the full dataset and the small subset in Table 10. Our observation shows that the optimal $\Delta$ selected based on the ratio results from the small subset is consistent with that selected based on the full dataset experiment results for the ISIC dataset. For Spambase, IMDB, and Criteo datasets, the optimal $\Delta$ selected based on the ratio results from the small subset is very close to that selected based on the full dataset in terms of the average ratio value. Figure 7 further visualizes this degree of closeness, where the three colored blocks on the full dataset represent the three sets of parameters with the smallest ratio. We find that the optimal parameter (red block) selected on the small subset is always included in the three sets of parameters with the smallest ratio on the full dataset.

Table 10: Average ratio results on the full dataset and small subset. For the ISIC dataset, the optimal $\Delta$ selected is consistent between the full dataset and 10% subset. Although for the Spambase, IMDB, and Criteo datasets, the optimal $\Delta$ selected differs between the full dataset and small subset, the selected ratio values are very close.

| dataset | sampling ratio | $\Delta = 0.05$ | $\Delta = 0.1$ | $\Delta = 0.2$ | $\Delta = 0.3$ | $\Delta = 0.5$ |
|---|---|---|---|---|---|---|
| Spambase | 1 | 0.727 | 0.733 | **0.722** | 0.723 | 0.746 |
| | 0.1 | **0.849** | 0.856 | 0.968 | 0.992 | 1.043 |
| IMDB | 1 | **0.627** | 0.648 | 0.895 | 0.833 | 0.833 |
| | 0.1 | 1.189 | **1.076** | 1.323 | 1.158 | 1.309 |
| Criteo | 1 | 1.061 | **0.941** | 1.226 | 0.993 | 1.011 |
| | 0.1 | 1.359 | 1.342 | 1.331 | 1.328 | **1.306** |
| ISIC | 1 | **0.933** | 0.955 | 0.990 | 1.009 | 1.022 |
| | 0.1 | **1.234** | 1.246 | 1.238 | 1.243 | 1.241 |

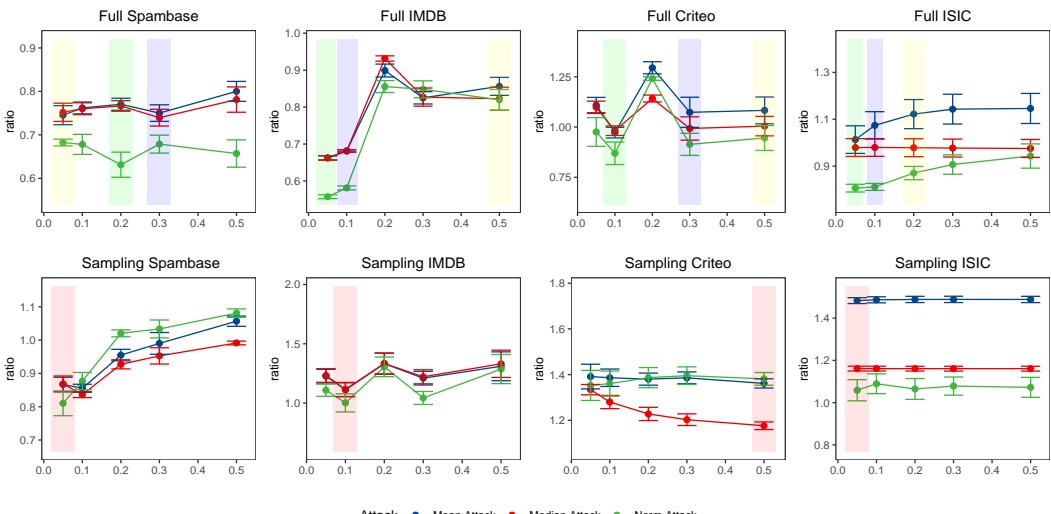

Figure 7: The ratio distribution on the full dataset and small subset. In Figure 7, the green (smallest), blue (second smallest), and orange (third smallest) blocks correspond to the three smallest ratio values on the full dataset for different $\Delta$, while the red block represents $\Delta$ with the smallest average ratio on the small subset. Notably, we observe that $\Delta$ with the smallest average ratio on the small subset always belongs to the three groups of $\Delta$ with the smallest average ratio on the full dataset.

# E    DATA SETUP AND EXPERIMENT DETAILS

In this section, we provide a detailed overview of our experimental setup. We first describe the pre-processing steps for the four public datasets in E.1. Next, we provide a description of the model architecture used for each dataset in E.2. Finally, we present the training hyperparameters for each dataset and model combination in E.3.

## E.1    DATASET PROCESSING

**Spambase** Spambase data are classified as whether they are spam or not. The Spambase dataset contains 4061 instances, each with 55 continuous real attributes and 2 continuous integer attributes. To preprocess the data, we replace all NA values with 0 and normalize the features to reduce the impact of magnitude differences. During each repetition, we split the dataset into a 70%-30% train-test split at different random seed settings.

**IMDB** The IMDB data has been preprocessed and words are encoded as a sequence of word indexes in the form of integers. We select the top 500 words and encode indexes with one-hot. The 50000 reviews are split into 25000 for training and 25000 for testing at different random seeds. And the final training and test data are both $25000 \times 500$ matrices filled with 0s and 1s.

**Criteo** The Criteo dataset consists of a portion of Criteo's traffic over a period of 7 days. Each row corresponds to a display ad served by Criteo. The dataset comprises 13 integer features and 26 categorical features, all of which have been hashed onto 32 bits. We followed the winner of the Criteo Competition's recommendations for data preprocessing[5] and the code[6] to process the entire dataset, rather than just a 10% subset like the Marvell paper. Our preprocessing steps included removing infrequent features that appeared in less than 10 instances, transforming numerical features (I1-I13) into categorical features, and treating them as a single feature. Additionally, we discretized numerical values using $log2$ transformation. For each iteration, we split the dataset into an 80%-20% train-test split using different random seed settings.

---

[5] https://www.csie.ntu.edu.tw/~r01922136/kaggle-2014-criteo.pdf
[6] https://github.com/rixwew/pytorch-fm/blob/master/torchfm/dataset/criteo.py

**ISIC** The official SIIM-ISIC Melanoma Classification dataset contains a total of 33126 skin lesion images, with less than 2% positive examples. To preprocess the ISIC data, we followed the approach outlined in this code[7]. During each iteration, we revise the image size $64 \times 64 \times 1$ and randomly split the dataset into an 80%-20% train-test split, using different random seed settings for each split.

## E.2  MODEL ARCHITECTURE DETAILS

To maintain consistency across all datasets, we utilized a 1-layer DNN as the generator and discriminator for our GAFM model. The intermediate results $f(\mathbf{x})$ were already transformed into linear embeddings, making them easily handled by the 1-layer DNN. For the generator and discriminator, we used LeakyReLU as the activation function for all layers except for the generator's output layer, which used the Sigmoid activation function. In the following sections, we will discuss the specific local model architectures used for feature extraction on each dataset.

**Spambase** For the Spambase dataset, the local model architecture consists of two linear layers with a LeakyReLU activation function. The first layer transforms the input size to a hidden representation of $batchsize \times 16$, and the second layer maps the hidden representation to a one-dimensional output using a sigmoid function.

**IMDB** The local model architecture for the IMDB datase is a fully connected model with 3 linear layers. The model consists of two hidden layers with 256 and 128 units, respectively, followed by ReLU activation functions and a dropout rate of 0.5. The output layer has a single unit after a sigmoid function.

**Criteo** In the context of online advertising data from Criteo, we utilize a widely-used deep learning model architecture called Wide&Deep model (Cheng et al., 2016). To implement Wide&Deep model, we refer to the code[8]. The architecture of Wide&Deep model consists of three main components: FeaturesLinear, FeaturesEmbedding, and MultiLayerPerceptron. The FeaturesLinear module is a linear model that uses embedding layers and a bias term to model feature interactions. The embeddings have a desired output dimension of 1. And the FeaturesEmbedding and MultiLayerPerceptron components take embedding layers as input features and uses a series of fully connected layers to learn complex feature interactions. In this implementation, the embedding dimension is set to 16 and the MLP dimension is set to $16 \times 16$. These components are combined in the local model, which adds a linear output layer with an output dimension of 1.

**ISIC** The local model for ISIC consists of 3 convolutional layers with 6, 16, and 32 output channels respectively. Each convolutional layer uses a $3 \times 3$ filter with a stride of $1 \times 1$. Following each convolutional layer is a ReLU activation function, and the output of each activation is max-pooled with a $2 \times 2$ window and stride size of $2 \times 2$. The output of the third convolutional layer is then flattened into size $32 \times 6 \times 6$ and passed into 3 fully connected layers, with 120, 84, and 40 units respectively.

## E.3  MODEL TRAINING DETAILS

We set the transformation function $F(.)$ to the identity function $I(.)$, clip the value at $c = 0.1$, use random seeds from 0 to 9. We employ PyTorch with CPU for Spambase and IMDB and employ PyTorch with CUDA acceleration to conduct all experiments for Criteo and ISIC. The Spambase dataset takes approximately 10 minutes per repetition, while the IMDB dataset requires around 0.5 hours. For the Criteo dataset, each repetition takes about 2.5 hours, and for the ISIC dataset, it takes about 40 minutes per repetition.

**Spambase, IMDB** We use the Adam optimizer with a batch size of 1028 and a learning rate of 1e-4 throughout the entire training of 300 epochs.

**Criteo** We use the Adam optimizer with a batch size of 256 and a learning rate of 1e-4 throughout the entire training of 100 epochs.

---

[7]https://github.com/OscarcarLi/label-protection/blob/main/preprocess_ISIC.ipynb

[8]https://github.com/BrandonCXY/Pytorch_RecommenderSystem/blob/master/DL%20Models%20Implementation%20in%20Recommender%20Sys%20with%20Pytorch.py

**ISIC** We use the Adam optimizer with a batch size of 256 and a learning rate of 1e-6 throughout the entire training of 250 epochs.

## F   THE COMBINATION OF GAFM AND MARVELL

In this section, we use the Criteo dataset as an example to demonstrate that combining GAFM and Marvell can enhance the privacy protection of GAFM. Table 11 shows the utility and privacy of GAFM, Marvell, and their combination (referred to as the G-M model) on the Criteo dataset, with the same experiment setup as the section 4.1. We observe a significant decrease in the leak AUC of the G-M model compared to Marvell and GAFM, demonstrating its improved ability to mitigate LLG.

Table 11: An example of the combination of GAFM and Marvell on Criteo. Table 11 demonstrates that the G-M model outperforms GAFM and Marvell in terms of lower leak AUC, indicating a better ability to protect label privacy, despite some utility degradation.

| Method | Utility | | | Privacy | | |
|---|---|---|---|---|---|---|
| | Avg. AUC | Worst AUC | Best AUC | Norm Attack | Mean Attack | Median Attack |
| GAFM | 0.67 | 0.64 | 0.73 | 0.68±0.06 | 0.80±0.09 | 0.77±0.05 |
| Marvell | 0.70 | 0.65 | 0.76 | 0.76±0.08 | 0.86±0.08 | 0.78±0.04 |
| G-M | 0.64 | 0.62 | 0.65 | 0.60±0.02 | 0.66±0.01 | 0.65±0.01 |

