# OpenReview forum: "GAN-based Vertical Federated Learning for Label Protection"
_ICLR.cc/2024/Conference — Submitted to ICLR 2024_

### Official Review · Reviewer_KHth · 2023-10-30

**Soundness:** 3 good
**Presentation:** 3 good
**Contribution:** 2 fair
**Rating:** 5
**Confidence:** 4

**Summary:**

The paper proposes a novel approach called the Generative Adversarial Federated Model (GAFM) to mitigate label leakage attacks from gradients in binary classification tasks in vertical federated learning. GAFM combines the vanilla splitNN architecture with Generative Adversarial Networks (GANs) to indirectly incorporate labels into the model training process. The GAN discriminator within GAFM allows federated participants to learn a prediction distribution that closely aligns with the label distribution, effectively preventing direct use of labels. Additionally, GAFM incorporates an additional randomized cross-entropy loss to enhance model utility. The paper evaluates the effectiveness of GAFM on various datasets and demonstrates its ability to mitigate label leakage without significant degradation in model utility.

**Strengths:**

- This paper combines GAN and splitNN to mitigate label leakage in vertical federated learning and conducts ablation experiments to investigate the role of each part.
- The paper is well organized and easy to follow.

**Weaknesses:**

- The author needs to provide a sufficient explanation for the ability of GAFM to mitigate label leakage. In section 3.4, the author claimed that GAFM has more mixed intermediate gradients but does not provide further explanation of the reasons for this phenomenon and the motivation for combining the GAN method.
- In Figure 3, both the GAN method and the CE method cause a decrease in model utility due to less label information leakage. However, their combination increases model AUC and leak AUC compared to the GAN-only model. It needs to be fully explained.
- GAFM does not provide sufficient privacy protection, as the attack AUC is mostly above 0.6, while the baseline Marvell in its original article can reduce the normal attack to 0.5.
- The paper needs to provide information on how to partition the features in its experiment. Different feature partitions will result in very different experimental results.

**Questions:**

- In this paper, why does Marvell perform much worse than the original paper? In the original Marvell paper, Norm Attack AUC can be reduced to 0.5 by Marvell on the same dataset, e.g., Criteo. Since the experimental results are not clearly explained, it is hard to tell the contributions made in the paper.

---

> ### Author Response · Authors · 2023-11-18
> **Response to Reviewer KHth**
>
> 【**Q1**】The author needs to provide a sufficient explanation for the ability of GAFM to mitigate label leakage. In section 3.4, the author claimed that GAFM has more mixed intermediate gradients but does not provide further explanation of the reasons for this phenomenon and the motivation for combining the GAN method.
>
> 【**A1**】Thanks for your question. We explained the mechanism behind GAFM's protection against label leakage in Section 3.4, Appendix B: Heuristic Justification for Improved Gradient Mixing, and Appendix C.1: Intermediate Gradients on Additional Datasets through experiments and analysis. We found that the protection arises from the mutual cancellation of GAN gradients and CE gradients. If you have other confusions regarding these analyses, we would be happy to provide further clarification.
>
> 【**Q2**】In Figure 3, both the GAN method and the CE method cause a decrease in model utility due to less label information leakage. However, their combination increases model AUC and leak AUC compared to the GAN-only model. It needs to be fully explained.
>
> 【**A2**】Sorry for the confusion regarding Figure 3. The CE loss does not cause a decrease in model utility, and there is no reduction in label information leakage for CE Loss. On the contrary, based on Figure 3 and Table 3, we can observe that the increased model utility of the CE loss is accompanied by poorer privacy protection.
>
> 【**Q3**】GAFM does not provide sufficient privacy protection, as the attack AUC is mostly above 0.6, while the baseline Marvell in its original article can reduce the normal attack to 0.5.
>
> 【**A3**】There is typically a trade-off between privacy protection and utility. As we emphasized in our paper, actually both GAFM and Marvell can sacrifice some utility for privacy protection, but GAFM exhibits greater stability. To better demonstrate the advantages of GAFM, we have depicted the utility-privacy trade-off curve of GAFM and Marvell on the Spambase and ISIC datasets (please refer to **Common Response [A2]**). From the results, we can observe that compared to Marvell, regardless of the Norm, Mean, or Median attack, GAFM achieves higher utility (test AUC) when privacy (leak AUC) is similar, and GAFM achieves lower privacy when utility is similar. Additionally, GAFM exhibits lower variance in its changes compared to Marvell, making it more stable.
>
> 【**Q4**】The paper needs to provide information on how to partition the features in its experiment. Different feature partitions will result in very different experimental results.
>
> 【**A4**】Please see the Model Architecture section (page 7) and Section C.2: Evaluation in the Multi-Party Setting, where we describe how we allocated features in our experiments for both the two-party setting and the multi-party setting.
>
> 【**Q5**】In this paper, why does Marvell perform much worse than the original paper? In the original Marvell paper, Norm Attack AUC can be reduced to 0.5 by Marvell on the same dataset, e.g., Criteo. Since the experimental results are not clearly explained, it is hard to tell the contributions made in the paper.
>
> 【**A5**】We apologize if our previous explanation was not clear. We would like to provide further clarification on our empirical experiments. The main message we aim to convey is that GAFM, as a novel gradient-perturbation framework that does not solely rely on random noise, demonstrates an improved utility-privacy trade-off compared to the vanilla splitNN structure and existing label protection methods such as Marvell and MaxNorm. In Table 2 of our paper, we demonstrated that GAFM achieves label protection comparable to Marvell under three attacks (norm, mean, median), while also achieving a slightly improved test AUC (utility) across repetitions with different random seeds. Furthermore, in Table 2 of our paper, we also observe GAFM exhibits greater stability compared to Marvell, which can also be observed through the Pareto curves for the utility-privacy trade-off, as presented in **Common Response [A2]**.
>
> It is worth noting that the privacy of Marvell is influenced by its parameter S. In the original paper, when S=4, Marvell indeed reduces the normal attack to 0.5, but it also leads to a decrease in model utility (as discussed in **Common Response [A1]**). Since we do not have information about the random seed and S used for each dataset in Marvell, we utilized the default parameter S=1 and our own random seeds in our paper. Based on our analysis of S in the **[Common Response [A1]**, we believe that S=1 is a reasonable choice that strikes a balance between privacy and utility for the ISIC and Spambase datasets.
>
> We hope that our clarification has addressed your concerns. If you have any further questions, we are more than happy to discuss them.

---

> > ### Comment · Reviewer_KHth · 2023-11-21
> >
> > I appreciate the authors for their efforts in updating the experiments and providing clarifications. I agree with the authors that the mutual cancellation of GAN gradients and CE gradients provides protection. However, I have two concerns:
> > 1. The motivation behind using the GAN-based method needs to be sufficiently justified, as Marvell and MaxNorm also generate mixed gradients as GAN does.
> > 2. According to the utility-privacy trade-off curve, the experimental results of Marvell still need to be revised between the original paper and this paper.
> >
> > Nevertheless, this paper represents an intriguing step towards improved label protection.

---

> ### Author Response · Authors · 2023-11-21
>
> Dear reviewer KHth,
>
> We appreciate your time in reviewing our paper! After carefully reviewing your comments, we have made clarifications by conducting additional experiments and providing further explanations. As the author-review discussion period is concluding, we kindly request your feedback on whether your previous concerns have been adequately addressed. We are also available to address any additional inquiries you may have. Your constructive feedback has been invaluable in improving the quality of our work, and we sincerely thank you for it.
>
> Authors

---

> ### Author Response · Authors · 2023-11-22
>
> Dear reviewer KHth,
>
> Thank you for your reply. We are glad that you agree with our viewpoint that GAFM provides protection through the mutual cancellation of GAN gradients and CE gradients, and think GAFM represents an intriguing step towards improved label protection. We also hope that the idea of GAFM which implicitly learns the target label distributions instead of labels can bring new inspiration to the community.
>
> As mentioned in the introduction (page 2), our motivation for using GANs is to avoid the issue of label leakage from gradients (LLG) caused by directly inferring sample labels from gradient calculations. The gradients here are obtained by directly involving sample labels in the computation. Instead, we use GANs to learn the distribution of labels in learning the instance class information while avoiding the LLG caused by the direct use of labels. Our experimental results (table 2) have indeed observed the privacy protection brought by this implicit utilization of labels. Additionally, we do not emphasize that GAFM is the state-of-the-art compared to Marvell, but rather highlight the larger variance that Marvell faces in terms of utility and privacy as the shown in the tabel 2 and Common Response [A1] [A2]. In the revised version, we will add the utility-privacy trade-off curve to better demonstrate the advantages of GAFM.
>
> We express our gratitude for the time and effort you have dedicated as the reviewer of our paper. Your valuable suggestions have significantly contributed to the improvement of our article.
>
> Authors

---

### Official Review · Reviewer_ph6x · 2023-10-31

**Soundness:** 2 fair
**Presentation:** 2 fair
**Contribution:** 2 fair
**Rating:** 3
**Confidence:** 5

**Summary:**

This paper focuses on the label leakage and protection in split learning (vertical federated learning) binary classification where the label-owning party wants to protect the label information from the other parties while allowing the joint learning of different parts of a predictive model. The authors propose a new approach by incorporating Generative Adversarial Networks in the model formulation: the label party learns a generator as the classifier and indirectly use the label information by matching the predicted label distribution rather than predicting the explicit labels (through the help of a learned discriminator). To further improve the performance, the author propose an additional cross-entropy loss to use the cut layer features to predict the example’s randomly perturbed labels to further boost performance. Experimentally, the authors show that the proposed method GAFM achieves a better model utility and privacy compared to existing baselines.

**Strengths:**

The paper focuses on solving an important practical problem of Label leakage in split learning.

**Weaknesses:**

- **Applying GAN to a binary classification seems unprincipled**. By using a GAN formulation which matches the predicted label distribution and the ground truth label distribution, the correspondence between an example’s feature $x$ and its label $y$ is completely lost. Suppose there are 50% positive examples and 50% negative examples. Then the target label distribution under the proposed gaussian additive perturbation would be an equal mixture of two Gassians $\mathcal{N}(0, \sigma^2)$ and $\mathcal{N}(1, \sigma^2)$. If the GAN is learned perfectly, we can only guarantee that the generator $G_\theta$ will also output this exact same Gaussian mixture (where the randomness is induced by the randomness over the example $x$). However, this cannot guarantee that all the positive $x$ (with $y=1$) will be mapped to $\mathcal{N}(1, \sigma^2)$ — instead, it is possible that the opposite can also happen, where all the positive examples are actually incorrectly mapped to $\mathcal{N}(0, \sigma^2)$ while the negative examples are all mapped to $\mathcal{N}(1, \sigma^2)$. In this case, the classifier would achieve an accuracy of $0$. Thus this current GAN formulation can’t guarantee the proper learning of a predictive generator classifier. Even if the authors additionally use the $L_{\textrm{CE}}$ to improve the performance, this additional loss cannot change the property of the generator $\theta_g$. Thus I don't believe this issue can be theoretically prevented.

- **Reliability of the experimental results**. The results presented in Table 2 makes me unsure whether the implementation and hyperparameters are tuned correctly for the experiments, thus making me question the reliability of the results:
    - **Too much variation in the AUC performance measured for Marvell**. In Table 2, the authors show that model utility (test AUC) for the method Marvel over 10 random seeds can have a range as wide as > 0.2 (on Spambase min=0.59, max=0.82). This is likely a result of incorrect code implementation or inappropriately-tuned learning rate. The authors explain this away by pointing out that Marvell adds Gaussian noise perturbation to intermediate gradients which can lead to fluctuating leak AUC values. However, this fluctuation is over the privacy metric (leak AUC) instead of the utility metric, and the fluctuation results from it being computed over a random training batch. However, the authors do not claim there is any noticeable variation over the utility metric (which is averaged over all test data) over different random seeds. This wide performance range of Marvell shown in this paper makes me concerned that the paper’s experimental results are not carefully checked and tuned.
    - **norm leak AUC of max_norm**. In the Marvell paper, the leak AUC under the norm attack of the method Max Norm is < 0.6 on Criteo (Figure 4) and around 0.7 on ISIC (Figure 9). However, in Table 2, the authors report a leak AUC number of 0.92 and 0.99 for Max Norm on Criteo and ISIC respectively. Considering max norm is a heuristic approach proposed specifically to protect against the norm attack, this much worse (higher is worse) result compared to that reported in the original paper makes me concerned the experimental evaluation are not fully correct.

- **Lack of privacy utility tradeoff visualization**. In Li et al (2022), the proposed method Marvell has a tuneable hyperparameter $s$ which controls the privacy-utility tradeoff: using higher values of $s$ allows more privacy but lower utility and this tradeoff frontier is visualized by Li et al. Thus it is expected for the authors to discuss whether their methods can achieve a privacy-utility tradeoff and if so present a complete tradeoff curve instead of showing only an individual point on the curve in Table 2.

- **Choice of attack methods**.
    - The authors mention on page 6 that the relationship between the sign and the cluster of gradients maybe not be as straightforward (and thus haven’t experimented with the cosine attack). However, this explanation is not clear and deserves more elaboration. Besides, regardless of the exact formulation of GAFM, the cosine leak AUC can still be evaluated and should be reported.
    - **Practicality of mean attack and median attack**. These two new attacks introduced by the authors assume the attackers know the gradient centers of class 0 and class 1. However, it’s not clear how the attackers can obtain this information in the first place. Computing the class specific mean requires taking the average over each corresponding class, thus requiring the knowledge of the class identities in the first place. Hence, the logic becomes circuitous. The authors should explain more clearly whether they believe these two attacks are realistic, and if so, how they can be achieved in practice.

- Typo: at the end of page 4, $\hat{y} = D_{\theta_g}(f(x))$ should be $\hat{y} = G_{\theta_g}(f(x))$.

**Questions:**

- Figure 2 is confusing. According to the caption, it is expected to be a visualization of the intermediate gradients colored by their ground truth label. However, each intermediate gradient should be a high-dimensional vector but is shown as a single number in Figure 2. Besides, it’s also not clear to me why the predictions should be shown on the horizontal axis for this figure. Can the authors provide an explanation of how to interpret the Figure?

---

> ### Author Response · Authors · 2023-11-18
> **Response to Reviewer ph6x**
>
> 【**Q1**】**Applying GAN to a binary classification seems unprincipled**
>
> 【**A1**】We want to clarify that while the phenomenon that the reviewer raised is fair, this is not an issue in our setting: the active party knows the true labels during the training. As long as the GAN can learn the distributions of different classes, even if the learned  distributions are completely opposite, such as mapping all positive instances to $\mathcal{N}(0,\sigma^2)$ and negative instances to $\mathcal{N}(1,\sigma^2)$. After training is completed, the active party can easily impose a rule of probability/label swapping to achieve a higher training AUC.
>
> 【**Q2**】**Too much variation in the AUC performance measured for Marvell**.
>
> 【**A2**】Thank you for your question.
>
> We would like to point out that Marvell did not report important random parameters such as random seed and precise experimental numbers. Therefore, our experimental results are based on our own parameters. We conducted re-experiments on the Spambase and ISIC datasets and indeed observed variance in the changes made by Marvell. For example, the test AUC of Spambase changed from 0.60 (seed=9) to 0.82 (seed=6). We have printed all the training details for your reference in the [`GAFM_Marvell_Spam_variance.ipynb`](https://anonymous.4open.science/r/Generative-Adversarial-Federated-Model-BFF7/GAFM_Marvell_Spam_Variance.ipynb) file.
>
> Additionally, we tried different values of S and found that the high variance in Marvell is not only a problem when S=1.  For instance, for the Spambase dataset, when S=0.25, the Marvell test AUC was only 0.3 (seed=2 in [`Curve_Spam.ipynb`](https://anonymous.4open.science/r/Generative-Adversarial-Federated-Model-BFF7/Curve_Spam.ipynb)). And the Pareto graph also demonstrates the larger variance of Marvell. For detailed results, please refer to the **Common Response** **[A2]**.
>
> We acknowledge that our speculation about the higher variance in Marvell potentially being due to the addition of noise may not be correct. We will further clarify this speculation in our article.
>
> 【**Q3**】**Norm leak AUC of max_norm**
>
> 【**A3**】Taking ISIC as an example, we once again observed the failure of MaxNorm in defending against norm attacks. In 250 epochs, it gradually decreased from 1 to 0.9. Please directly refer to the file [`ISIC_MaxNorm.ipynb`](https://anonymous.4open.science/r/Generative-Adversarial-Federated-Model-BFF7/ISIC_MaxNorm.ipynb) for more details.
>
> Meanwhile, it is worth noting that our experiments are not the only ones that have observed the failure of MaxNorm defense against Norm Attack. In this [experiment](https://github.com/Koukyosyumei/Attack_SplitNN/blob/main/examples/Label_Leakage.ipynb), the researchers also observed that MaxNorm was ineffective in defending against Norm Attack on the Credit Card Fraud Detection dataset, with a (flipped) leak AUC of 0.92.
>
> 【**Q4**】 **Lack of privacy utility trade-off visualization.**
>
> 【**A4**】Thank you for your suggestion. We have incorporated the privacy utility trade-off on the spambase and ISIC datasets. Please refer to the **Common Response** **[A2]** for more details. We will include these experimental results in the revised version.
>
> 【**Q5**】The authors mention on page 6 that the relationship between the sign and the cluster of gradients maybe not be as straightforward (and thus haven’t experimented with the cosine attack). However, this explanation is not clear and deserves more elaboration. Besides, regardless of the exact formulation of GAFM, the cosine leak AUC can still be evaluated and should be reported.
>
> 【**A5**】Cosine attack depends on which labeled samples available and the mean/median attacks are more like a stabilized version of it, and also doe assume any sign. If the norm is 1, then $cos(\theta_1, \theta_2)=1-0.5\||\theta_1-\theta_2\||^2_2$, $\theta_1$ is mean/median vector rather than one sample in positive or negative class  and $\theta_2$ is the one sample under investigation. In addition, the results in Table 2 of the paper indeed demonstrate that the Mean/Median Attack is a threatening attack method. Thank you for your suggestion. We also plan to add results of the cosine attack in the revised version.
>
> 【**Q6**】**Practicality of mean attack and median attack**.
>
> 【**A6**】Thanks for your question. Like Marvell, we define an oracle attack that has access to more accurate and strong information, such as the gradient mean/median of class 0 and class 1 in the mean/median attack, and the clean gradient of one positive example in the cosine attack. In this case, any protection method capable of defending against this oracle attack would also protect against the more practical one (the same attack without this accurate and strong information).
>
> 【**Q7**】The typo at the end of page 4.
>
> 【**A7**】Thank you for pointing it out. We have fixed this typo.

---

> > ### Author Response · Authors · 2023-11-18
> > **Response to Reviewer ph6x**
> >
> > 【**Q8**】Figure 2 is confusing. According to the caption, it is expected to be a visualization of the intermediate gradients colored by their ground truth label. However, each intermediate gradient should be a high-dimensional vector but is shown as a single number in Figure 2. Besides, it’s also not clear to me why the predictions should be shown on the horizontal axis for this figure. Can the authors provide an explanation of how to interpret the Figure?
> >
> > 【**A8**】To better visualize the variations in intermediate gradients across different samples and the introduction of label information through CE loss, we directly consider the simplest structure where the cut layer $f(x)$ is aggregated into a 1-dimensional representation. Consequently, the intermediate gradient becomes extremely simple, as it is merely a scalar value. We would like to emphasize that if GAFM can guarantee mutual cancellation of GAN Loss and CE Loss in the aggregated one-dimensional $f(x)$, it can also guarantee mutual cancellation in high dimension because for each dimension $f_i(x)$, according to the chain rule, we have $\frac{\partial Loss}{\partial f_i(x)} = \frac{\partial Loss}{\partial f(x)}\frac{\partial f(x)}{\partial f_i(x)} \propto \frac{\partial Loss}{\partial f(x)}$, where $f(x)=\frac{1}{d}\sum^d_i f_i(X)$.
> >
> > In Figure 2, the x-axis represents the predicted values of the instances, the y-axis represents the corresponding gradients, and the color represents the true labels of the instances. The x-axis reflects the utility of GAFM as it represents the predicted values. The y-axis visually demonstrates the differences between GAN gradients and CE gradients, as well as how their fusion leads to more mixed gradients.
> >
> > We appreciate your suggestions, and we are also happy to answer any other questions you may have.

---

> > ### Comment · Reviewer_ph6x · 2023-11-22
> > **Response to author rebuttal I**
> >
> > I would like to thank the authors for their rebuttal. However, my current position on the paper’s decision remains unchanged, as I believe the proposed methods are not principled and the experimental evaluations are incorrect.
> >
> > - **Unprincipledness of applying GAN in classification**. The authors have acknowledged my point that by using a GAN loss to match the marginal label predictive distribution and the ground truth marginal label distribution, it is possible that the generator could map the positive examples all to the negative label and the negative examples to the positive label. The authors claim that in this case, the label party could just flip the sign of the generator as a hack to address this issue. Although this hack can work for this special case (which I was only giving as an example), it wouldn’t work for other cases: consider the case where we still have 50% positive examples and 50% negative examples, so the smoothed ground truth label marginal distribution is still an equal mixture of $\mathcal{N}(0, \sigma^2)$ and $\mathcal{N}(1, \sigma^2)$. Now suppose there is a generator that maps the distribution of the positive examples to the exact same mixture, while also mapping the distribution of the negative examples to same mixture as well. In this case, the predicted label’s marginal distribution (the average of the same mixtures is still that mixture) is exactly the same as the ground truth marginal distribution, so this generator would have the minimal possible loss under the GAN objective. However, the test AUC under this gan classifier would be 0.5 (which is equivalent to random guessing) and flipping the sign of this GAN classifier would not help, as it gives out the same AUC of 0.5. This is not nitpicking to find an edge case. What I’m trying to point out is that the GAN objective is proposed for distribution matching on the __marginal distribution level__ and the authors are also using it to match two label’s marginal distributions. However, in classification, we care a lot more than matching the label’s marginal distributions — we actually want to match the learned model’s __conditional label distribution__ given $x$ to the ground truth conditional label distribution given the same $x$ (oftentimes as a one-hot distribution in the training data) for all possible $x$. Thus I believe fundamentally the GAN formulation are inappropriate for a classification problem in the way it is used here. This is very likely also the reason that GAN only performs so poorly as pointed out by Reviewer vM2y.

---

> > > ### Comment · Reviewer_ph6x · 2023-11-22
> > > **Response to author rebuttal II**
> > >
> > > - **Incorrect computation of utility metrics**. After looking a further look at the code in `GAFM_Marvell_Spam_Variance.ipynb` in the anonymized link, I believe the author is computing the evaluation metrics incorrectly. There, the authors compute the train and test AUC by having the model make predictions on each training and test batch, and then concatenating all the predictions values and comparing it against the ground truths labels. What is incorrectly done here is that the authors are updating the model parameters across batches, so the predictions do not belong to the same model parameters. This practice is commonly done to track training accuracies and training cross entropies as a cheap way to avoid looping over the training dataset again. In those cases, this practice is more acceptable as the metrics computed are example-level — the cross entropy on one example has no effect on the cross entropy on another example. However, this is far from the case when computing AUC, which is a set level metric. As the metrics are computed not using the same model parameters across training updates, the metrics could be significantly miscalculated from the actual real value if the authors were to run the evaluation over the dataset without any model parameter change.
> > > - **Computing the privacy metrics.** The authors have also used the same aggregation approach to compute the leak AUCs (the same issue could also happen), while the authors of Marvell have only computed these metrics over each individual batch and report the 95% quantiles over all the batch level leak AUC values seen during training. I suggest the authors use the same way of computing the privacy metrics for consistent comparisons.
> > > - **Are the optimization done correctly for Marvell?** In the `GAFM_Marvell_Spam_Variance.ipynb` notebook cell 66 and 58, the training losses do not decrease much at all during the 300 epochs of training (for example, 0.002703 at epoch 0 to 0.002689 at epoch 299 for seed 6). This makes me wonder if the authors have done any learning rate tuning for Marvell. As Marvell adds random perturbations to the intermediate gradients, the resulted parameter gradients will have higher variance than normal training. Thus it might be necessary to use lower learning rate as we see in this example the training losses are not decreasing much for Marvell. Can the authors report how the learning rate is tuned for Marvell? A lower learning rate can potentially make the parameter learning a lot more stable and also improve Marvell’s utility (at no cost to privacy).

---

> ### Author Response · Authors · 2023-11-21
>
> Dear Reviewer ph6x,
>
> Thank you once again for your valuable feedback on our paper. We have taken your comments into serious consideration and conducted additional experiments to address the raised questions. As the author-review discussion period is reaching its conclusion, we kindly request your feedback on whether your previous concerns have been satisfactorily resolved.  Given the time constraint, please let us know if there are any other questions or concerns you would like us to address. We highly appreciate your support and truly value your guidance in improving the quality of our work. Thanks!
>
> Authors

---

> ### Author Response · Authors · 2023-11-22
> **Response to "Unprincipledness of applying GAN in classification".**
>
> We thank the reviewer for raising this question and are happy to clarify it. In our case, the input f(x) to the GAN structure is also the input of the CE loss, which encourages f(x) to be close to a noisy version of the true y and, consequently, encourages closeness in the GAN's output given the limited learning complexity in our GAN structure.
>
> Considering the example that the reviewer is describing: Suppose we have (z1, ...., zn) from N(0, sigma^2), and (x1,...., xn) from N(1, sigma^2), and we use (z1,...., zn), (x1,...,xn) as our inputs to the GAN, discriminating them against 1/2N(0, sigma^2) + 1/2 N(1, sigma^2). The shallow and narrow GAN initialized at the identity is expected to maintain the majority of the relative distances among the points to achieve a low loss. To a certain extent, our GAN's output is conditional on the input f(x). Previous work has employed a similar idea for cross-modality or cross-condition mapping and has achieved good empirical results. For example, in [1], the authors transformed gene expression data into images by applying GAN to the embedded latent space, and in [2], the authors transformed observations from one batch to another using GAN.
>
> We agree with the reviewer on the importance of being cautious when relying on GANs for prediction (the potential risks apply to both label alignment and multi-dimensional settings). It is crucial to use relevant input and restrict the learning capacity of the GAN by employing a simple structure or regularizations.
>
> [1] Yang, Karren Dai, et al. "Multi-domain translation between single-cell imaging and sequencing data using autoencoders." Nature communications 12.1 (2021): 31.
>
> [2] Wang, Yuge, Tianyu Liu, and Hongyu Zhao. "ResPAN: a powerful batch correction model for scRNA-seq data through residual adversarial networks." Bioinformatics 38.16 (2022): 3942-3949.

---

> ### Author Response · Authors · 2023-11-22
> **Respones to comments on utility/privacy metrics.**
>
> Again, we thank the reviewer for the valuable comments!
>
> [Incorrect computation of utility metrics] We agree that it is better to calculate test AUC using the final trained model. The test AUC results are comparable to the old results, with the differences depending on the data sets. For example, for Marvel, the re-calculated test AUCs for Spambase are [0.761, 0.630, 0.808, 0.726, 0.684, 0.837, 0.760, 0.718, 0.521, 0.577] using seeds 0-9, with a mean test AUC being 0.70 (previously 0.71); the re-calculated test AUCs for ISIC are [0.707,0.773,0.723,0.6621,0.556,0.660,0.731,0.708,0.761,0.671], with a mean test AUC being 0.69 (previously 0.64). We will update our test AUC results using the final model in the revised manuscript.
>
> [Computing the privacy metrics] We currently used the mean of leak AUC from individual batches instead of 95% percentile because we believe the mean is more stable and equally accounts for all samples without further human interference (e.g., why 95%, not 80% or 50%). However, we do agree that the same measure of in Marvel can help with the comparisons across manuscripts and will add the new evaluations in our revised supplement.
>
> [Are the optimization done correctly for Marvell] We did not specifically adjust the learning rate parameters and have used a a fixed learning rate of 1e-4 for both vanilla splitNN, Marvel, and GAFM in the spambase example. We are currently retraining Marvell model with 10% (1e-5) of currently used learning rate to examine if it has a huge impact.

---

### Official Review · Reviewer_wbsE · 2023-11-01

**Soundness:** 3 good
**Presentation:** 3 good
**Contribution:** 2 fair
**Rating:** 6
**Confidence:** 2

**Summary:**

This paper proposes a GAN-based approach to transform the upload embedings for prediction from non-label party. In this way, the backpropagated gradients are perturbated. This prevents label leakage from gradients during model training. The empirical results demonstrate the effectiveness of GAFM among four public datasets, Spambase, IMDB, Criteo, ISIC. Furthermore, the authors also investigate which component (e.g., GAN or cross-entropy) is necessary to make GAFM work.

**Strengths:**

1. The empirical results are promising.
2. The paper is well-organized and easy to follow.

**Weaknesses:**

1. This paper only investigates a specific kind of vertical federated learning. In the setting, the authors assume that all the participants without labels share the same model architecture, which is not so practical. Besides, they only investigate on the binary tasks.
2. The GAN-based approach to protect labels seems similar to [1]. They seem to prevent the attacks by transforming the uploaded embedding. This paper may be improved by adding some comparative experiments or some analysis in related work.

[1] Scheliga, Daniel, Patrick Mäder, and Marco Seeland. "Precode-a generic model extension to prevent deep gradient leakage." Proceedings of the IEEE/CVF Winter Conference on Applications of Computer Vision. 2022.

**Questions:**

1. what is the model architecture of the label party? What would be the experiment comparison between the different approaches in the context of the label party owning many layers of a neural network?

---

> ### Author Response · Authors · 2023-11-18
> **Response to Reviewer wbsE**
>
> 【**Q1**】This paper only investigates a specific kind of vertical federated learning. In the setting, the authors assume that all the participants without labels share the same model architecture, which is not so practical. Besides, they only investigate on the binary tasks.
>
> 【**A1**】Thank you for your comment. We want to emphasize that GAFM does not impose a constraint that all participants without labels must share the same model architecture. Whether the gradients between GAN and CE loss perturb each other is not directly related to the local model architectures of each unlabeled party.
>
> Regarding binary tasks, as mentioned in the Introduction section, binary tasks often face a severe Label Leakage Gradient (LLG) problem due to the use of cross-entropy loss, resulting in significant gradient differences between positive and negative samples. However, limited research has been conducted to address this issue. Moreover, binary classification is widely used in various federated scenarios, such as healthcare, finance, credit risk, and smart cities. Therefore, in this paper, similar to our baselines (Marvell and MaxNorm), we also focus on binary classification tasks. We agree with your viewpoint that exploring the extension of GAFM to multi-class problems is important. We will consider it as a direction for future work, although it is beyond the scope of this paper.
>
> 【**Q2**】The GAN-based approach to protect labels seems similar to [1]. They seem to prevent the attacks by transforming the uploaded embedding. This paper may be improved by adding some comparative experiments or some analysis in related work.
>
> 【**A2**】Thank you for suggesting this very interesting work which also utilizes generative models for protecting the data privacy. We will include this paper in the related work section. However, it is worth noting that although both our work and the referenced paper [1] employ the idea of perturbing gradients for gradient protection by generative models, we address different problems. Our focus is on preserving the privacy of labels rather than preventing the leakage of training data. Additionally, our problem scenario specifically revolves around vertical federated learning.
>
> 【**Q3**】what is the model architecture of the label party? What would be the experiment comparison between the different approaches in the context of the label party owning many layers of a neural network?
>
> 【**A3**】Thanks for your question. We provide the details of the model architecture of the "label party" in Table 1 and the Model Architecture section. We found that 1-layer DNNs are sufficient as the generator and discriminator since their inputs are very simple.
>
> We also provide additional experimental results using the Spambase dataset as an example, considering the label party with varying numbers of layers (from 1 to 3) in a neural network. We observed that the number of layers has almost no impact on the results.
>
> - **Experiments on Spambase across 10 repeats.**  We observed that the number of layers had little to no effect on GAFM.
>
> | Model   | Test AUC         | Norm Attack     | Mean Attack     | Median Attack   |
> | ------- | ---------------- | --------------- | --------------- | --------------- |
> | 1-Layer | 0.93             | 0.56 $\pm$ 0.04 | 0.67 $\pm$ 0.05 | 0.66 $\pm$ 0.05 |
> | 2-Layer | 0.93  $\pm$ 0.02 | 0.56 $\pm$ 0.04 | 0.67 $\pm$ 0.05 | 0.66 $\pm$ 0.05 |
> | 3-Layer | 0.93 $\pm$ 0.07  | 0.56 $\pm$ 0.05 | 0.67 $\pm$ 0.06 | 0.66 $\pm$ 0.06 |
>
> Thank you for your suggestion. We hope the above clarification answers your question. If you have other questions, we are happy to answer them.

---

> ### Author Response · Authors · 2023-11-21
>
> Dear reviewer wbsE,
>
> Thank you for dedicating your time to review our paper and for your support of our work! We have thoroughly reviewed your comments and made necessary adjustments by conducting additional experiments and providing further explanations. As the author-review discussion period is coming to a close, we kindly ask for your feedback on whether your previous concerns have been adequately addressed. We are also here to address any further inquiries you may have. We sincerely appreciate your constructive feedback, as it has contributed to enhancing the quality of our work. Thank you!
>
> Authors

---

### Official Review · Reviewer_vM2y · 2023-11-01

**Soundness:** 2 fair
**Presentation:** 2 fair
**Contribution:** 2 fair
**Rating:** 6
**Confidence:** 4

**Summary:**

This paper introduces a GAN-based Federated Model, GAFM, to address the binary label leakage problem in VFL. GAFM employs GANs as a surrogate to indirectly utilize label information, thereby mitigating LLG. To reduce its negative impact on model performance, an additional cross-entropy loss is further adopted to enhance prediction accuracy. Empirical results on various datasets demonstrate that GAFM achieves a better balance between model utility and privacy.

**Strengths:**

- **Tackling an important problem in VFL**: LLG is a crucial security issue in VFL, impacting many significant applications. Achieving an optimal balance between utility and privacy is the ultimate goal in this field.

- **Effective Method with Insightful Explaination**:  Backpropagating the GAFM loss results in a blend of two gradients whose distribution centers diverge in opposite directions. Coupled with Proposition 3.1, the paper effectively explains this mutual gradient perturbation of GAN and CE losses as the reason for GAFM's success. Empirical demonstration in Figure 2 and theoretical analysis in Appendix B provide a comprehensive understanding of the effectiveness of the proposed method.

**Weaknesses:**

**Unclarities in Experiment**
- The value of the **hyperparameter** $s$ significantly affects the performance **of Marvell**, yet it is not mentioned in this paper. The details of training and evaluating Marvell should be provided.
- It would be more insightful to display a **utility-privacy trade-off curve** for a more intuitive performance comparison. (e.g., varying $s$ for Marvel to generate its dot-curve and varying $\Delta$ to produce GAFM's dot-curve, as shown in Figure 4 of the Marvell paper)
- As depicted in Figure 3, 'GAN only' demonstrates extremely poor performance on Criteo and IMDB, with AUC values close to 0.5. It appears that the GAN fails to learn the label distribution, thereby losing its purpose as a surrogate, which contradicts its initial motivation . From my perspective, the generator should, to some extent, successfully simulate the distribution, but it now seems that the performance of the generator is not crucial. **This leads me to question the necessity of GAN**. Perhaps any other method that provides the opposite gradient center with respect to CE loss could replace the GAN module. I hope the authors can provide a reasonable explanation for this.

**Unclarity in Method Design**
- What is the underlying intuition behind designing the CE loss to measure the distance between $f(x)$ and the randomized response $\tilde y$? Given that $f(x)\in R^d$ is a hidden vector and $\tilde y \in R$ is a noisy label, why is it reasonable or necessary to reduce the dimension of $f(x)$ to 1 by averaging? Would using a single FC layer be an alternative to achieve this purpose? A clear explanation is needed here.

**Presentation need improvement**
- Highlighting the best performance in the table would help readers grasp the key point quickly.
- Figure 1 is quite misleading when it comes to understanding the dimension of $\hat y$ and the structure of the discriminator. In fact, for the binary classification problem, its structure is extremely simple as the input is merely a scalar value. Since the primary real-world applications of VFL are in tabular data, using MLP for model structure demonstration would be smoother.

**Questions:**

As stated in the Weakness part.

---

> ### Author Response · Authors · 2023-11-18
> **Response to Reviewer vM2y**
>
> 【**Q1**】The value of the **hyperparameter S** significantly affects the performance **of Marvell**, yet it is not mentioned in this paper. The details of training and evaluating Marvell should be provided.
>
> 【**A1**】Thank you for your valuable suggestion. In our experiments, we uniformly used the default value of $S=1$ in Marvell due to the lack of detailed descriptions regarding the use of parameter $S$ for each dataset. However, we acknowledge your perspective on parameter $S$. As a result, we supplemented the impact of hyperparameter $S$ on Marvell for both Spambase and ISIC in our numerical experiments. Please refer to the **Common Response 【**A1**】** for more details.
>
> We observed that increasing $S$ improves the privacy protection of Marvell but simultaneously leads to a decrease in performance. For Spambase and ISIC, $S=1$ is a reasonable choice that balances performance and privacy protection. Further adjusting the parameter to control the privacy protection capability of Marvell around 0.5 will result in a more noticeable difference between utility and GAFM. However, it does not alleviate the instability issue of Marvell.
>
> 【**Q2**】It would be more insightful to display a **utility-privacy trade-off curve** for a more intuitive performance comparison. (e.g., varying S for Marvel to generate its dot-curve and varying Δ to produce GAFM's dot-curve, as shown in Figure 4 of the Marvell paper)
>
> 【**A2**】Thank you for your suggestion. We also demonstrated the utility-privacy trade-off curve for both Marvell and GAFM on Spambase and ISIC datasets. Please refer to the conclusion provided in **Common Response 【**A2**】**. In summary, we observed that across the Norm Attack, Mean Attack, and Median Attack scenarios, GAFM consistently exhibits a better trade-off curve than Marvell. When the test AUC is similar, GAFM consistently achieves a lower leak AUC than Marvell, and Marvell's leak AUC exhibits significant variance.
>
> 【**Q3**】 As depicted in Figure 3, 'GAN only' demonstrates extremely poor performance on Criteo and IMDB.. Perhaps any other method that provides the opposite gradient center with respect to CE loss could replace the GAN module. I hope the authors can provide a reasonable explanation for this.
>
> 【**A3**】 We agree with your point that methods other than GANs can potentially be utilized to cancel the gradients of the original CE loss. The challenge lies in finding a method that cancels the gradients effectively while minimizing the loss in utility. The advantage of using GANs is that they provide a reasonable loss function that allows us to learn the target distribution. Additionally, we have observed that even in cases where a GAN-only model fails to generate accurate predictions by aligning the label distribution, it does not significantly harm the overall prediction performance.
>
> 【**Q4**】What is the underlying intuition behind designing the CE loss to measure the distance between *f*(*x*) and the randomized response ? Why is it reasonable or necessary to reduce the dimension of *f*(*x*) to 1 by averaging? Would using a single FC layer be an alternative to achieve this purpose? A clear explanation is needed here.
>
> 【**A4**】For the 'underlying intuition of f(x)', f(x) represents the distribution generated jointly by the unlabeled participants. The active party with labels transforms this joint distribution using the generator and discriminator into a predictive distribution that closely matches the label distribution. We introduce the CE loss to incorporate more label information, aiming to make the predictive distribution obtained from f(x) more accurately reflect the label distribution. Our ablation study (4.2.2 ABLATION STUDY) also supports our viewpoint that using only the GAN loss sometimes leads to less accurate predictions, while the CE loss can introduce more label information to improve accuracy.
>
> Regarding 'reduce the dimension of f(x) to 1', the design in the paper is not the only option. We used the simplest method of gradient aggregation, as it allows for visualization of the gradients' change and facilitates the convenient introduction of label information through the CE loss. However, an appropriate single FC layer can also be used to achieve this objective, and the one-dimensional f(x) obtained through summation can be seen as a single FC layer with equal weights. We want to emphasize that if GAFM can guarantee mutual cancellation of GAN Loss and CE Loss in the aggregated one-dimensional f(x), it can also guarantee mutual cancellation in each dimension $f_i(x)$. This is because, according to the chain rule, $\frac{\partial Loss}{\partial f_i(x)} = \frac{\partial Loss}{\partial f(x)}\frac{\partial f(x)}{\partial f_i(x)} \propto \frac{\partial Loss}{\partial f(x)}$, where $f(x)=\frac{1}{d}\sum^d_i f_i(X)$.

---

> > ### Author Response · Authors · 2023-11-18
> > **Response to Reviewer vM2y**
> >
> > 【**Q5 and Q6**】Highlighting the best performance in the table would help readers grasp the key point quickly.
> > Figure 1 is quite misleading when it comes to understanding the dimension of  and the structure of the discriminator. In fact, for the binary classification problem, its structure is extremely simple as the input is merely a scalar value. Since the primary real-world applications of VFL are in tabular data, using MLP for model structure demonstration would be smoother.
> >
> > 【**A5 and A6**】Thank you for your suggestion. In the revised version, we will highlight the best results in Table 2, and we will also make the correction in Figure 1 by replacing the existing CNN structure with an MLP structure.
> >
> > Thank you once again for your constructive feedback. We hope that these clarifications can address your concerns, and if you have any further questions, we would be happy to answer them.

---

> ### Author Response · Authors · 2023-11-21
>
> Dear reviewer vM2y,
>
> Thank you again for taking the time to review our paper and for your support of our work! We have carefully considered your comments and incorporated additional experiments and explanations. As the author-review discussion period is nearing its end, we kindly request your feedback on whether your previous concerns have been adequately addressed. We are also available to address any further questions you may have. We appreciate your constructive feedback, which has greatly contributed to improving the quality of our work. Thank you!
>
> Authors

---

### Author Response · Authors · 2023-11-18
**Common Response**

We sincerely appreciate the comments and constructive suggestions from the four reviewers. We are glad that the reviewers consider our research topic is important (Reviewer vM2y, Reviewer ph6x) and find the proposed methods is effective with insightful explanations (Reviewer vM2y, Reviewer wbsE). The reviewers also praised the well-organized and easy-to-follow writing of the paper (Reviewer wbsE, Reviewer KHth). We would also like to thank the reviewers for suggesting and incorporating additional evaluations and experiments that have helped improve the understanding of GAFM.

Before addressing each question raised point by point, we would like to present additional experiment results using different tuning parameters for Marvell and our proposal, along with the associated Pareto curve (utility-privacy trade-off curve) for test AUC and leak AUC. This suggestion was made by Reviewer vM2y and Reviewer ph6x, and it can also help answer the question raised by Reviewer ph6x and Reviewer KHth regarding why Marvell results in our paper appear to have a discrepancy compared to the original paper.

We use the Spambase and ISIC datasets as examples to demonstrate the influence of parameter S on Marvell (Response **【A1】** ). Based on these results, we plot the Utility-Privacy Trade-off Curve for Marvell and GAFM  (Response **【A2】** ).

**【A1】**  Based on the original Marvell paper, we adjusted the parameter range of $S$ to include values of 0.1, 0.25, 1, and 4, each repeated three times. Our experimental results are consistent with the Marvell paper, indicating that increasing the parameter $S$ can enhance privacy protection capability (achieving leak AUC close to 0.5). However, it is important to note that this increase in privacy protection comes at the cost of a decrease in utility (test AUC).

 - **Tuning hyperparameter S on Spambase**. The smaller average test AUC for S = 0.25 is due to an outlier in Marvell's test AUC when seed = 2, which resulted in a test AUC of 0.33.

  | S    | Avg. AUC   | Norm Attack     | Mean Attack     | Median Attack   |
  | ---- | --------------- | --------------- | --------------- | --------------- |
  | 0.1  | 0.91 $\pm$ 0.02 | 0.76 $\pm$ 0.09 | 0.93 $\pm$ 0.05 | 0.91 $\pm$ 0.01 |
  | 0.25 | 0.69 $\pm$ 0.25 | 0.64 $\pm$ 0.16 | 0.88 $\pm$ 0.06 | 0.85 $\pm$ 0.04 |
  | 1    | 0.81 $\pm$ 0.10 | 0.55 $\pm$ 0.05 | 0.78 $\pm$ 0.12 | 0.76 $\pm$ 0.10 |
  | 4    | 0.76 $\pm$ 0.06 | 0.53 $\pm$ 0.03 | 0.68 $\pm$ 0.07 | 0.68 $\pm$ 0.06 |

  - **Tuning hyperparameter S on ISIC**

| S    | Avg. AUC   | Norm Attack    | Mean Attack    | Median Attack  |
| ---- | --------------- | -------------- | -------------- | -------------- |
| 0.1  | 0.84 $\pm$ 0.01 | 0.80$\pm$ 0.10 | 0.86$\pm$ 0.10 | 0.84$\pm$ 0.09 |
| 0.25 | 0.75$\pm$ 0.03  | 0.75$\pm$ 0.02 | 0.82$\pm$ 0.05 | 0.81$\pm$ 0.05 |
| 1    | 0.65 $\pm$ 0.01 | 0.65$\pm$ 0.02 | 0.70$\pm$ 0.00 | 0.68$\pm$ 0.01 |
| 4    | 0.57$\pm$ 0.03  | 0.51$\pm$ 0.01 | 0.64$\pm$ 0.06 | 0.64$\pm$ 0.06 |

**【A2】**  Pareto curve ( utility-privacy trade-off curve) for test AUC and leak AUC

We used the results of Marvell at different $S$ values from the table above to characterize the Pareto curve for test AUC and leak AUC. Additionally, for GAFM, we adjusted the parameter range of $\Delta$ to include values of 0.05, 0.1, 0.2, 0.3, and 0.5 (the experimental results are directly from Appendix D.3 DISCUSSION ON $\Delta$).

We then plotted the Test AUC-leak AUC curves for Marvell and GAFM separately under different attacks in the [Utility-Privacy Trade-off Curve_Spam_ISIC.pdf](https://anonymous.4open.science/r/Generative-Adversarial-Federated-Model-BFF7/Utility-Privacy%20Trade-off%20Curve_Spam_ISIC.pdf). We observed that, when facing three types of attacks, GAFM exhibits a better trade-off between utility and privacy compared to Marvell. Specifically, when the test AUC is similar, GAFM has a lower leak AUC, and when the leak AUC is similar, GAFM has a larger test AUC. Additionally, we noticed significant variation in Marvell's leak AUC.

Again, we sincerely appreciate the reviewers’ suggestions on how to improve our paper, and we hope with our revision and responses, we can successfully demonstrate that GAFM can improve privacy protection via non-random perturbation, utilizing a new insight that the gradients of CE loss and GAN loss naturally tend to have opposite trends for two classes.

---

### Meta-Review · Area_Chair_ZCNM · 2023-12-10

**Metareview:**

This paper proposed Generative Adversarial Federated Model (GAFM), a GAN-based approach to improve label privacy protection. The paper has received extensive discussions during the review period. Overall, the paper is below the bar of ICLR.

Strength:
1. The presentation is clear.

Weakness:
1. Motivations using GAN for the task is not clear. Thus the novelty of the paper is in question.
2. Experiments are likely to contain wrong and unrealistic settings.

**Justification For Why Not Higher Score:**

The paper has limited novelty and experimental results are not convincing.

**Justification For Why Not Lower Score:**

N/A

---

### Decision · Program_Chairs · 2024-01-16

Reject